EMBO
Molecular Medicine

# Trans-generational epigenetic regulation associated with the amelioration of Duchenne Muscular Dystrophy

Julie Martone[1,2,†], Michela Lisi[1,†], Francesco Castagnetti[1,3,†], Alessandro Rosa[1,4] iD, Valerio Di Carlo[5], Enrique Blanco[5], Adriano Setti[1], Davide Mariani[3], Alessio Colantoni[1], Tiziana Santini[4], Lucia Perone[6], Luciano Di Croce[5,7,8] iD & Irene Bozzoni[1,2,3,4,*] iD

## Abstract

Exon skipping is an effective strategy for the treatment of many Duchenne Muscular Dystrophy (DMD) mutations. Natural exon skipping observed in several DMD cases can help in identifying novel therapeutic tools. Here, we show a DMD study case where the lack of a splicing factor (Celf2a), which results in exon skipping and dystrophin rescue, is due to a maternally inherited trans-generational epigenetic silencing. We found that the study case and his mother express a repressive long non-coding RNA, DUXAP8, whose presence correlates with silencing of the Celf2a coding region. We also demonstrate that DUXAP8 expression is lost upon cell reprogramming and that, upon induction of iPSCs into myoblasts, Celf2a expression is recovered leading to the loss of exon skipping and loss of dystrophin synthesis. Finally, CRISPR/Cas9 inactivation of the splicing factor Celf2a was proven to ameliorate the pathological state in other DMD backgrounds establishing Celf2a ablation or inactivation as a novel therapeutic approach for the treatment of Duchenne Muscular Dystrophy.

**Keywords** ATAC sequencing; DMD Exon skipping; iPSC Reprogramming; lncRNAs; trans-generational epigenetic inheritance
**Subject Categories** Chromatin, Transcription & Genomics; Genetics, Gene Therapy & Genetic Disease; Musculoskeletal System

## Introduction

With a few exceptions, most DMD mutations lead to frameshifts, resulting in the formation of premature termination codons that impair dystrophin mRNA translation (Monaco *et al*, 1988). Exon skipping and rescue of a correct reading frame have been demonstrated to be effective for many mutations, potentially allowing the conversion of severe Duchenne genotypes into milder Becker phenotypes (Bladen *et al*, 2015; Verhaart & Aartsma-Rus, 2019). Due to the large size of introns, splicing enhancers play an important role in dystrophin mRNA maturation, and indeed, their binding sequences have been used as targets for effective antisense strategies to induce exon skipping (Incitti *et al*, 2010; Niks & Aartsma-Rus, 2017). Antisense oligonucleotides, by pairing with exonic splicing enhancers (ESEs), interfere with recognition by the splicing machinery and effectively induce exon skipping and rescue of dystrophin synthesis (Mitrpant *et al*, 2009). We previously reported the case of a DMD individual (GSΔ44), carrying the deletion of exon 44, who showed a milder phenotype resembling the Becker one due to spontaneous skipping of exon 45 coupled to 7% of dystrophin rescue. Such endogenous exon skipping was due to the lack of Celf2a (Martone *et al*, 2016), an isoform of the Celf2 splicing factor (Ladd *et al*, 2001).

Here, we show that the absence of Celf2a expression in the GSΔ44 patient is due to a trans-generational epigenetic silencing that is inherited from the mother and requires the activity of a repressive long non-coding RNA (lncRNA), DUXAP8. We found that Celf2a expression resumed when induced Pluripotent Stem Cells (iPSCs), deriving from the patient's fibroblasts, were differentiated into myocytes and that, in these conditions, exon 45 skipping was abrogated. Moreover, we demonstrate that Celf2a activation correlated with the knock down of DUXAP8; indeed, depletion of DUXAP8 in

1  Department of Biology and Biotechnology 'Charles Darwin', Sapienza University of Rome, Rome, Italy
2  CNR Institute of Molecular Biology and Pathology (IBPM), Rome, Italy
3  Center for Human Technologies, Istituto Italiano di Tecnologia, Genova, Italy
4  Center for Life Nano Science@Sapienza, Istituto Italiano di Tecnologia, Rome, Italy
5  Center for Genomic Regulation, Barcelona, Spain
6  Cell Culture and Cytogenetics Core, Telethon Institute of Genetics and Medicine, Pozzuoli, Italy
7  Universitat Pompeu Fabra (UPF), Barcelona, Spain
8  Institucio Catalana de Recerca i Estudis Avançats (ICREA), Barcelona, Spain
   *Corresponding author (lead contact). Tel: +39 06 4991 2202; E-mail: irene.bozzoni@uniroma1.it
   †These authors contributed equally to this work

GSΔ44 myoblasts rescued Celf2a expression, while its overexpression in control myoblasts repressed it. We also show that inactivation of Celf2a could have a more general therapeutic application since CRISPR/Cas9 inactivation of this factor in an unrelated Δ44 genetic background resulted in exon 45 skipping and partial rescue of dystrophin synthesis.

# Results

## Celf2a inactivation through CRISPR/Cas9 in Δ44 myocytes induces skipping of exon 45

Figure 1A shows that the lack of the splicing factor Celf2a in GSΔ44 myocytes correlates with skipping of exon 45 from the dystrophin mRNA (Martone *et al*, 2016). This correlation is further supported by the lack of exon skipping when Celf2a is exogenously expressed in GSΔ44 myocytes (Martone *et al*, 2016). To ascertain Celf2a function in different genetic backgrounds and whether its depletion alone could be therapeutic for other Δ44 mutations, we applied CRISPR/Cas9 (Li *et al*, 2016) targeted genome editing to specifically delete the Celf2a isoform in iPSCs derived from a different Δ44 patient (UPΔ44).

Fibroblasts from a skin biopsy of UPΔ44 were reprogrammed into iPSCs (Somers *et al*, 2010; Sommer *et al*, 2012). After reprogramming, single iPSC-like colonies were selected and expanded as individual clones that grew with the typical morphology of pluripotent stem cells (Fig EV1A). UPΔ44 clones were validated as bona fide iPSCs by qPCR assessment of pluripotency marker expression (NANOG, OCT4, SOX2, REX1) and downregulation of the direct inhibitor of OCT4 (NR2F2; Fig EV1B). The exogenous OCT4 silencing was also assessed (Fig EV1C). WT#1 clone has been previously validated as iPSC clone in Lenzi *et al* (2015). Cytogenetic analysis showed a normal karyotype of WT#1 and UPΔ44#3 clones (Fig EV1D). The genetic identity of these clones was assessed by PCR on genomic DNA, which confirmed the specific lack of exon 44 in UPΔ44#3 clone (a representative is shown in Fig EV1E).

Clone UPΔ44#3 was transfected with constructs expressing Cas9 and guide RNAs, together with a donor plasmid carrying a selection cassette flanked by homology arms, to delete the first exon specific for the Celf2a isoform (Fig EV1F). The parental cell line (UPΔ44#3) and two homozygous edited clones (UPΔ44#3.1 and UPΔ44#3.5) were then induced to muscle differentiation through enhanced piggyBac vector (epB)-mediated overexpression of MyoD and Baf60c (Lenzi *et al*, 2016). All clones showed a fusion index comparable to the differentiated WT iPSCs (Fig 1B and C) and similar expression of muscle markers indicating the occurrence of correct myogenic differentiation (Fig EV1G). A specific transcriptional knock out of the Celf2a isoform was obtained in the edited clones, while the expression of the other Celf2 isoforms (Celf2b and Celf2c) remained unaffected (Fig 1D). Interestingly, the absence of Celf2a in the edited clones correlated with the appearance of a band of the dystrophin transcript corresponding to the fusion of exon 43 with exon 46 (Fig 1D). Sequencing of this band confirmed the perfect skipping of exon 45 (Fig EV1H). Notably, under these conditions the rescue of the dystrophin protein was obtained (Fig 1E). Western blot indicated

that rescue was approximately 5% of the normal expression levels, similarly to what was observed in the biopsy and primary myoblasts of the GSΔ44 patient who has a mild Duchenne symptomatology (Martone *et al*, 2016). These results allowed to conclude that ablation of Celf2a is likely sufficient for the rescue of dystrophin production in DMD genetic backgrounds where skipping of exon 45 could restore a functional open reading frame (ORF).

## Celf2a repression in GSΔ44 is due to epigenetic silencing

iPSCs were also derived from the fibroblasts of GSΔ44 and his mother (GSM). The obtained clones (GSΔ44#2, GSΔ44#8 and GSM#1) were validated as bona fide iPSCs both morphologically (Fig EV2A) and molecularly (Fig EV2B and C). Cytogenetic analysis of GSΔ44#2 and GSΔ44#8 iPSC clones showed a normal karyotype (Fig EV2D). The genetic identity of these clones was analysed by PCR on genomic DNA, which confirmed the presence of the GSΔ44-specific deletion (Fig EV2E). Since the Celf2a isoform is specifically expressed in muscle cells, these clones, together with a control iPSC line, were differentiated into myocytes through the overexpression of MyoD (Lenzi *et al*, 2016) and Baf60c (Albini *et al*, 2013) and correct expression of muscle differentiation markers was confirmed (Fig EV2F). As opposed to GSΔ44 primary myoblasts which lack Celf2a (Fig 1A), in myocytes differentiated from GSΔ44 iPSCs (Myo, GSΔ44#2 and GSΔ44#8), a clear recovery of the expression of Celf2a was observed (Fig 2A, see in comparison with Fig 1A). Notably, in all clones, the expression levels of the Celf2b and 2c isoforms remained unaltered. In parallel to Celf2a reactivation, in myocytes derived from GSΔ44#2 and GSΔ44#8 exon 45 skipping of the dystrophin mRNA was abolished (Fig 2D). Similarly to GSΔ44, the expression of Celf2a, which was absent in myocytes obtained by direct trans-differentiation of fibroblasts from the mother (Martone *et al*, 2016), was instead resumed in myocytes derived from her iPSCs (Fig 2A, GSM#1).

To exclude the possibility that Celf2a expression could be mediated by MyoD overexpression or by the combined overexpression of Baf60c and MyoD, the epB vectors were transfected in GSΔ44 primary myoblasts: under these conditions, Celf2a expression was not recovered (Fig EV2G).

Altogether these data indicate, in line with the observation that the Celf2a gene of GSΔ44 does not show any obvious mutation (Martone *et al*, 2016), that the lack of Celf2a expression in GSΔ44 is due to epigenetic silencing and that this regulation is lost upon the reprogramming process.

To further strengthen the correlation between the lack of Celf2a and exon 45 skipping, we applied CRISPR/Cas9 to GSΔ44 iPSCs (clone GSΔ44#8) in order to specifically inactivate the Celf2a transcript. Several clones were analysed for editing, and one of them displayed a homozygous deletion of the first Celf2a exon (GSΔ44#8.2; Fig EV2H). In GSΔ44#8.2 clone, Celf2a expression was abolished while editing did not affect neither Celf2b nor Celf2c expression (Fig 2C). The ability of this clone to differentiate into myotubes upon the overexpression of Baf60 and MyoD, with the same extent of WT iPSCs, was confirmed by MHC immunostaining and fusion index analyses (Fig 2B and C). Notably, when analysing the dystrophin mRNA structure by RT–PCR (Fig 2D) or by sequencing (Fig EV2I), we observed a band corresponding to exon 45 skipping; furthermore,

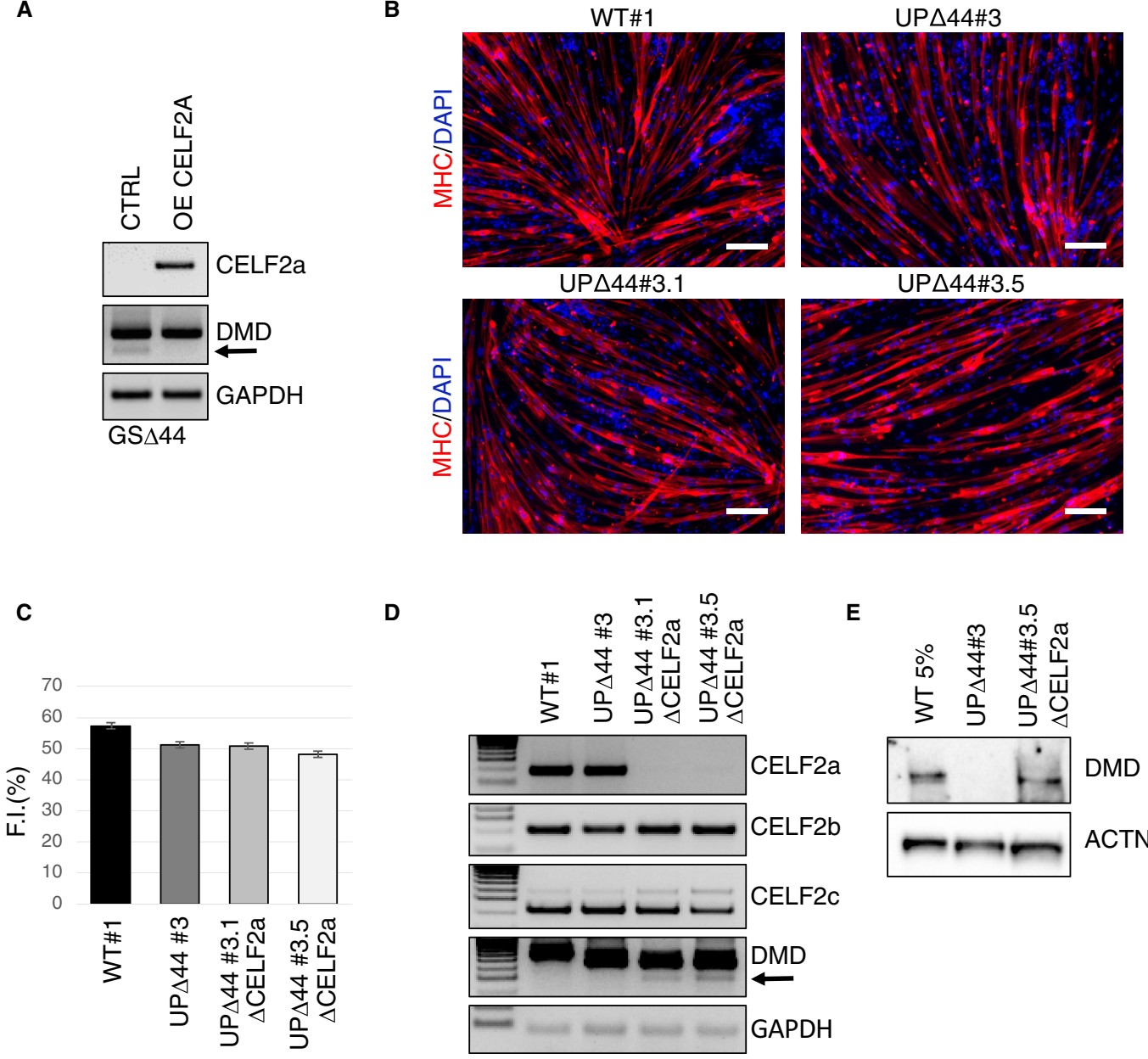

**Figure 1. Celf2a inactivation through CRISPR/Cas9 in UPΔ44 myocytes induces skipping of exon 45.**

A  GSΔ44 myoblasts were transfected with an empty vector (CTRL) or with a CELF2a overexpressing construct (OE CELF2A) (Martone *et al*, 2016). Cells were collected after 10 days in differentiation medium (DM), and the RNA was analysed by RT–PCR for CELF2a expression and DMD exon 45 skipping. GAPDH was used as loading control. The arrow indicates the DMD exon 45-skipped band. Representative results are shown (*n* = 3).

B  Representative immunofluorescence for myosin heavy chain (MHC in red) in combination with DAPI staining (in blue) of iPSCs obtained from WT (WT#1), UPΔ44 (UPΔ44#3) and Δ44 edited clones (UPΔ44#3.1 and UPΔ44#3.5) differentiated for 9 days into myocytes by MYOD/BAF60c overexpression. Scale bar 100 μm.

C  Histogram represents fusion index quantification. At least 7 randomly chosen microscope fields of two independent biological samples were analysed (*n* = 2). Data are presented as mean ± SD of the biological replicates.

D  RNA extracted from myocytes obtained by differentiation of WT (WT#1), UPΔ44 (UPΔ44#3) and edited UPΔ44 iPSC (UPΔ44#3.1 ΔCELF2a and UPΔ44#3.5 ΔCELF2a) clones was analysed by RT–PCR for CELF2a, CELF2b, CELF2c and DMD exon 45 skipping. GAPDH was used as loading control. The arrow indicates the DMD exon 45-skipped band. Representative results are shown (*n* = 3).

E  Western blot on proteins (40 μg) extracted from myocytes obtained by differentiation of WT (WT#1), UPΔ44 (UPΔ44#3) and edited UPΔ44 iPSC (UPΔ44#3.1 ΔCELF2a and UPΔ44#3.5 ΔCELF2a) clones probed with antibodies against dystrophin (DMD). Actinin (ACTN) was used as a loading control. For the WT 5% sample, 2 μg of proteins was diluted in 38 μg of proteins belonging to UPΔ44#3 to reach a total amount of 40 μg. Representative results are shown (*n* = 3).

Source data are available online for this figure.

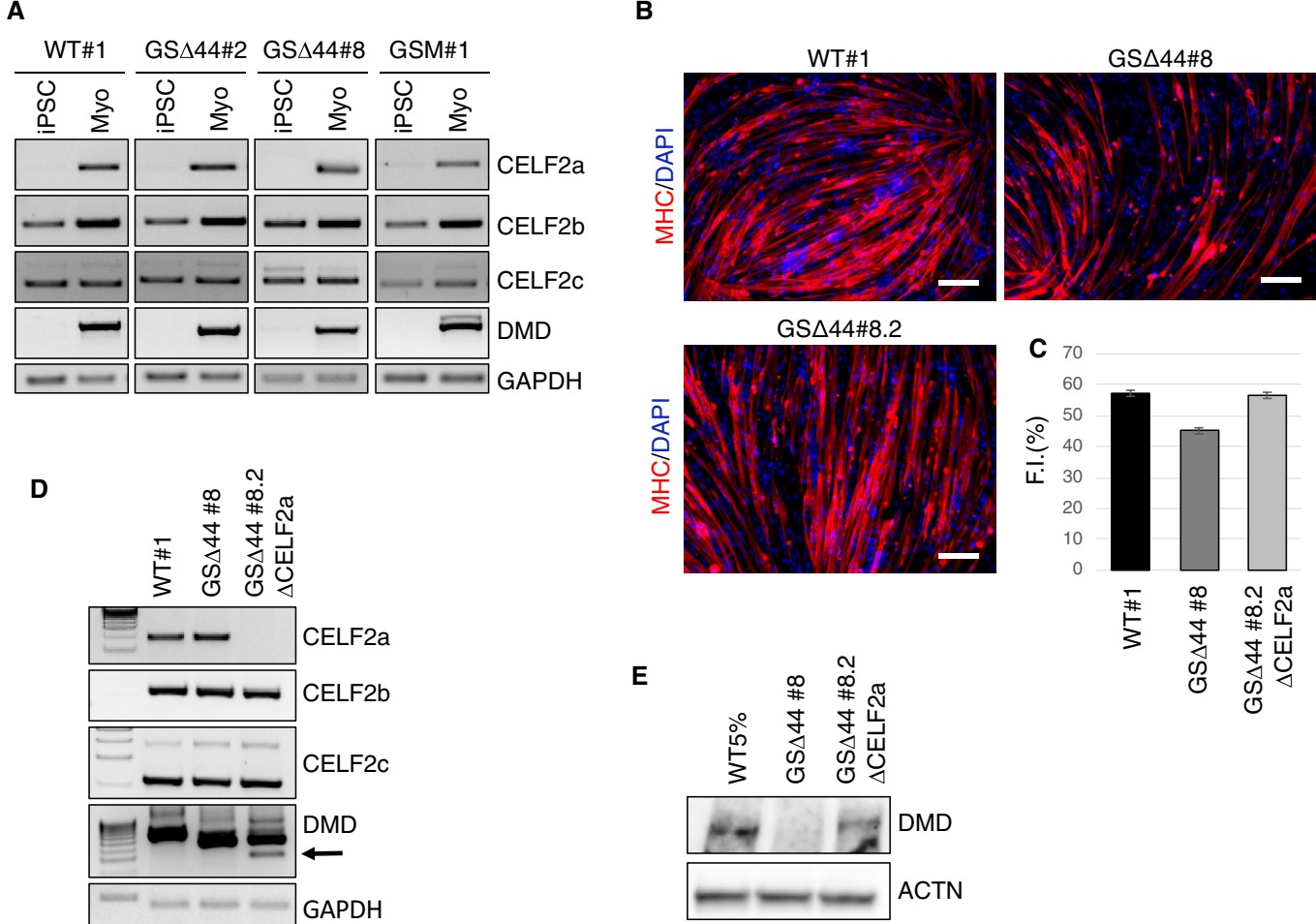

**Figure 2. Celf2a repression in GSΔ44 is due to epigenetic silencing.**

A   iPSCs obtained from a control (WT#1; Lenzi et al, 2015), GSΔ44 (GSΔ44#2, GSΔ44#8) and GSM (GSM#1) were differentiated into myocytes by MYOD/BAF60c overexpression. Cells were collected before the induction (iPSC) and after 9 days in DM (Myo), and RNA was analysed by RT–PCR for CELF2a, CELF2b, CELF2c and DMD expression. GAPDH was used as loading control. Representative results are shown (n = 3).

B   Representative immunofluorescence for myosin heavy chain (MHC in red) in combination with DAPI staining (in blue) of iPSCs obtained from WT#1, GSΔ44 (GSΔ44#8) and GSΔ44 edited clones (GSΔ44#8.2) differentiated for 9 days into myocytes by MYOD/BAF60c overexpression. Scale bar 100 μm.

C   Histogram represents fusion index quantification. At least 7 randomly chosen microscope fields of two independent biological samples were analysed (n = 2). Data are presented as mean ± SD of the biological replicates.

D   RNA extracted from myocytes obtained by differentiation of WT#1, GSΔ44 (GSΔ44#8) and GSΔ44-edited clones (GSΔ44#8.2 ΔCELF2a) was analysed by RT–PCR for CELF2a, CELF2b, CELF2c and DMD exon 45 skipping. GAPDH was used as loading control. The arrow indicates the DMD exon 45-skipped band. Representative results are shown (n = 3).

E   Western blot on proteins (40 μg) extracted from myocytes obtained by differentiation of WT (WT#1), GSΔ44 (GSΔ44#8) and edited GSΔ44 (GSΔ44#8.2 ΔCELF2a) iPSC clones probed with antibodies against dystrophin (DMD). Actinin (ACTN) was used as a loading control. For the WT 5% sample, 2 μg of proteins was diluted in 38 μg of proteins belonging from GSΔ44#8 to reach the total amount of 40 μg. Representative results are shown (n = 3).

Source data are available online for this figure.

under these conditions, rescue of the dystrophin protein was obtained (Fig 2E). Also in this case, Western blot indicated a rescue of approximately 5% of the normal expression levels, in line with what observed in the GSΔ44 patient (Martone *et al*, 2016).

In conclusion, we show that the lack of the Celf2a factor in the GSΔ44 case is due to epigenetic repression. Reactivation of the gene can be achieved by cell reprogramming followed by induction of myogenic differentiation that ensures proper tissue-specific expression of the gene.

## Chromatin accessibility and epigenetic signature in control and GSΔ44 myoblasts

Given the importance of the epigenetic silencing in regulating Celf2a expression in GSΔ44 cells, we decided to deepen our understanding of the chromatin accessibility by performing ATAC-seq analysis on GSΔ44 and healthy donor myoblasts (Fig 3A). The number of peaks called in the two conditions was comparable: 125,029 in control myoblasts (WT) and 128,349 in GSΔ44 (Fig 3B). Notably, we did

not observe genome-wide changes in peak distribution, suggesting no major chromatin rearrangements between the two conditions (Fig 3B, correlation Fig EV3A, fragment profiles Fig EV3B). To refine these data, we analysed all the peaks with a fold change higher than 2 and with an absolute value larger than 2 units in at least one of the two samples. We identified 505 upregulated and 1,007 downregulated peaks in GSΔ44 cells compared to control cells. Interestingly, a striking difference was observed in the Celf2

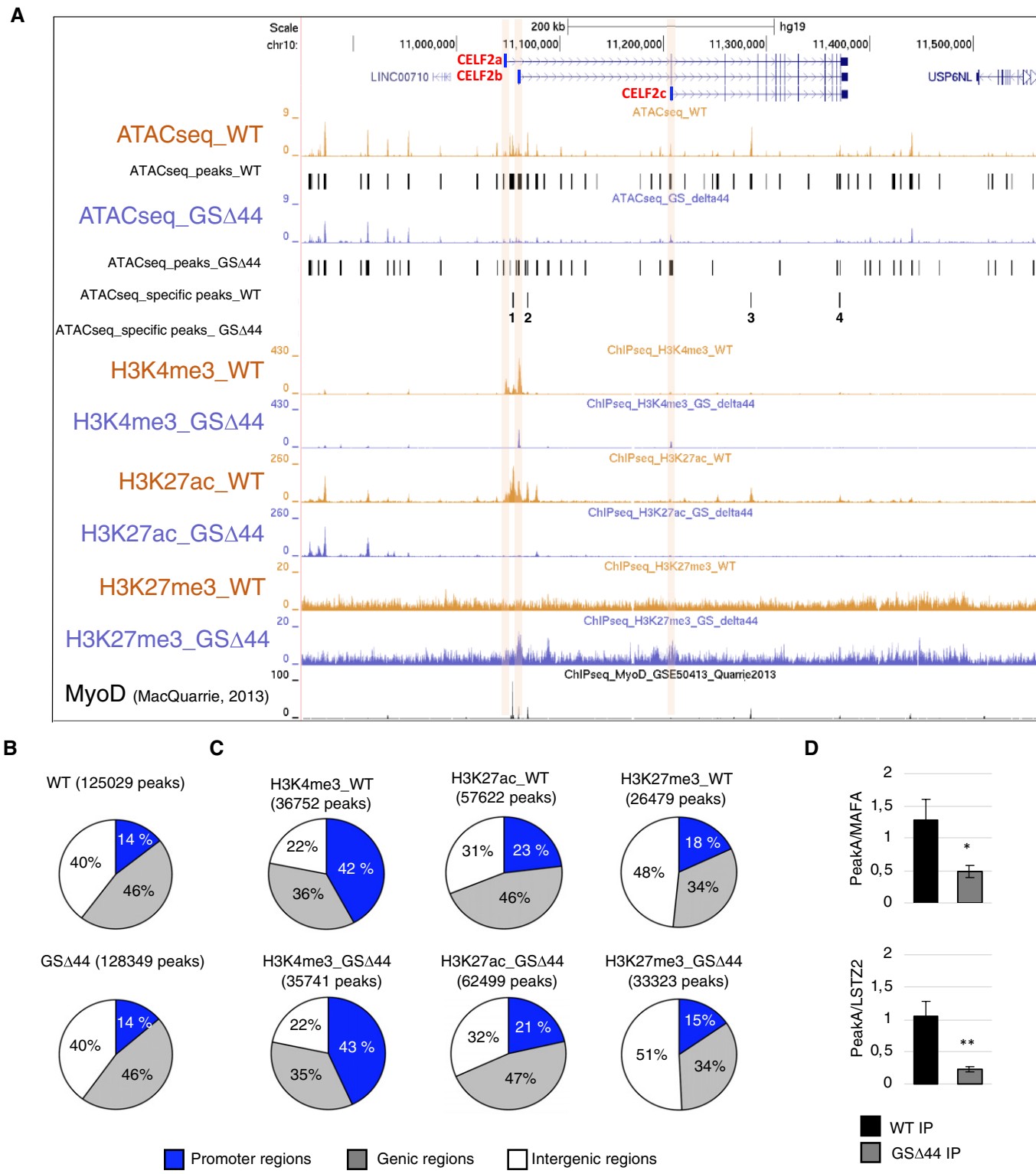

Figure 3.

◀

**Figure 3. Chromatin accessibility and epigenetic signature of control and GSΔ44 myoblasts.**

A   DNA accessibility (ATAC sequencing) and chromatin marks (ChIP-seq) signal in the Celf2 region (± 50 kb) using the UCSC genome browser. ATAC-seq peaks called in the two conditions are reported below each track. The differential peaks specific for one of the two conditions are reported in the tracks ATACseq_specific peaks_WT and ATACseq_specific peaks_GSΔ44. ChIP-seq signals for H3K4me3, H3K27ac and H3K27me3 in WT and GSΔ44 myoblasts are reported (WT myoblasts in light orange; GSΔ44 myoblasts in light blue). ChIP-seq for MyoD in control cells was obtained from MacQuarrie *et al* (2013) (GSM1218849). Orange boxes highlight the regions containing the first exon of each isoform (a–c).

B   Pie charts showing ATAC-seq peaks' genomic distribution (blue: promoter region, grey: gene bodies, white: intergenic regions).

C   Pie charts showing genomic distribution of ChIP-seq peaks (H3K4me3, H3K27ac and H3K27me3) (blue: promoter region, grey: gene bodies, white: intergenic regions).

D   Chromatins extracted from control (WT) and GSΔ44 myoblasts were immunoprecipitated with anti-MyoD and control IgG antibodies. Regions corresponding to the MyoD binding sites present in Peak A (see Fig EV3D) and in the MAFA and LSTZ2 genes were analysed by qPCR. Fold enrichments of PeakA over MAFA or LSTZ2 immunoprecipitation (IP) are plotted for WT and GSΔ44 samples. The mean ± SEM of triplicates from one representative experiment is shown (*n* = 3). Statistical significance of differences between means was assessed by a two-tailed *t*-test, and *P* < 0.05 was considered significant. \**P* = 0.0339, \*\**P* = 0.0067.

locus (± 50 kb), with a reduction in the total number of the called peaks from 33 in control condition to 24 in GSΔ44 cells. Four of these peaks, specifically present in the WT sample (Fig 3A, ATAC-seq, WT track, peaks 1–4), were absent in the GSΔ44 sample (Fig 3A, ATAC-seq, GSΔ44 track). The reduction in the number of ATAC-seq peaks, and the absence of upregulated ones, suggested an overall compaction of the chromatin in this region.

To strengthen our results, we performed ChIP-seq experiments for H3K4me3, H3K27ac and H3K27me3 in order to respectively identify active promoters, active enhancer regions and repressed chromatin (Fig 3A). The genome-wide analysis of peak number was comparable between WT and GSΔ44 for H3K4me3 and H3K27ac, whereas a moderate increase for H3K27me3 was observed in GSΔ44 cells (25,070 peaks in WT versus 31,852 peaks in GSΔ44). Moreover, the genomic distribution of the peaks for the three histone modifications was not affected genome-wide (Fig 3C). Interestingly, in GSΔ44 H3K4me3 marks could not be detected in correspondence of the Celf2a Transcription Start Site (TSS), while they were present at the TSS of the Celf2b and Celf2c isoforms. These data are in agreement with the lack of Celf2a expression and with the sustained production of the Celf2b and Celf2c mRNAs in GSΔ44 cells, respectively (Fig EV3C). When looking at four specific ATAC-seq peaks, present in the WT sample (Fig 3A, ATAC-seq, WT track, peaks 1–4), and absent in GSΔ44 (Fig 3A, ATAC-seq, GSΔ44 track), we found that they overlapped with H3K27ac and MyoD peaks (MacQuarrie *et al*, 2013 and Fig EV3D) while there was no correspondence with those of H3K4me3. Moreover, the loss of H3K27ac peaks in GSΔ44 was paralleled by gain of H3K27me3 modifications, well correlating with the chromatin compaction of the corresponding regions and with the lack of Celf2a expression.

In order to validate the differential accessibility of MyoD to the identified regions, MyoD-ChIP experiments were performed on GSΔ44 versus WT myoblasts. The first peak (A, Fig EV3D) was analysed by qPCR; two different positive controls (MAFA and LSTZ2) were used to normalize the IP efficiency in the two samples. An enrichment of almost threefold was observed in the WT sample with respect to GSΔ44 (Fig 3D), indicating that MyoD binding is impaired in GSΔ44 myoblasts. It remains to be established if the absence of MyoD on this site is the cause or the consequence of the closed chromatin state observed by ATAC-seq in GSΔ44 myoblasts.

Finally, it is of note the fact that the chromatin compaction observed in GSΔ44 does not affect Celf2b and c expression. Since these two mRNAs have a more ubiquitous expression, as indicated by their presence also in proliferating iPSCs, it can be derived that the four ATAC peaks identified in the Celf2 locus are likely to correspond to myogenic-specific chromatin structures.

## A trans-acting lncRNA controls Celf2a expression

The Celf2 locus spans more than 300 kb and contains, besides the a, b and c isoforms, also other two annotated Celf2-like transcriptional units (Celf2α and β, Fig 4A). Moreover, several lncRNAs are embedded in this locus (Fig EV4A), and in particular, a divergent lncRNA (SFTA1P) is transcribed in the opposite direction at a distance of 2.2 kb from the TSS of Celf2α. When looking at the expression of these genes in control, Δ44 and GSΔ44 myocytes, we noticed that the downregulation of Celf2a correlated with the lack of the SFTA1P lncRNA expression and with the activation of Celf2α and Celf2β (Figs 4B and EV4B). Notably, in myocytes obtained from differentiated GSΔ44 iPSCs, we observed that SFTA1P followed the same reactivation as Celf2a (Fig EV4F). Interestingly, the presence of SFTA1P expression in iPSCs indicated that epigenetic remodelling of the locus was indeed already occurring during the reprogramming process; instead, the lack of Celf2a expression in iPSCs could only be ascribed to the absence of the transcriptional factors needed for its muscle-specific expression.

SFTA1P initially appeared to be a promising candidate as a cis-acting regulator, because of its correlation with Celf2a expression; however, neither its overexpression in GSΔ44 myoblasts resulted in a rescue of Celf2a expression (Fig EV4C), nor was its downregulation in control myoblasts able to silence Celf2a expression (Fig EV4D). Moreover, the overexpression of Celf2a in GSΔ44 myoblasts had no effect on SFTA1P (Fig EV4E). These data indicated that there is no reciprocal regulation of the divergent transcripts SFTA1P and Celf2a, suggesting the existence of a trans-acting factor controlling both genes.

Therefore, we re-analysed total RNA-Seq data (Martone *et al*, 2016) from control, GSΔ44 and unrelated Δ44 myoblasts in order to search for alterations of the transcriptome possibly associated with Celf2a repression, focusing on non-coding genes. Through differential expression analysis, we identified 4,996 loci deregulated between GSΔ44 and control myoblasts and 2,843 between Δ44 and control (*q*-value < 0.05). Out of these, 219 and 100, respectively, resulted non-coding genes and 59 were deregulated in both conditions (Fig EV4G). A further selection produced a list of 9 lncRNAs expressed in GSΔ44, and neither in Δ44 myoblasts nor in control ones (Table EV1). Of these, only one was previously described to be associated with chromatin (DUXAP8, Ma *et al*, 2017).

We were able to validate the expression of this lncRNA in GSΔ44 myoblasts (GM) and myocytes (DM) and its absence in control cells (Fig 4C). Moreover, in myoblasts, DUXAP8 resulted to be equally distributed between cytoplasm and chromatin fractions (Fig EV4H). In all cases, the expression of DUXAP8 and Celf2a was mutually

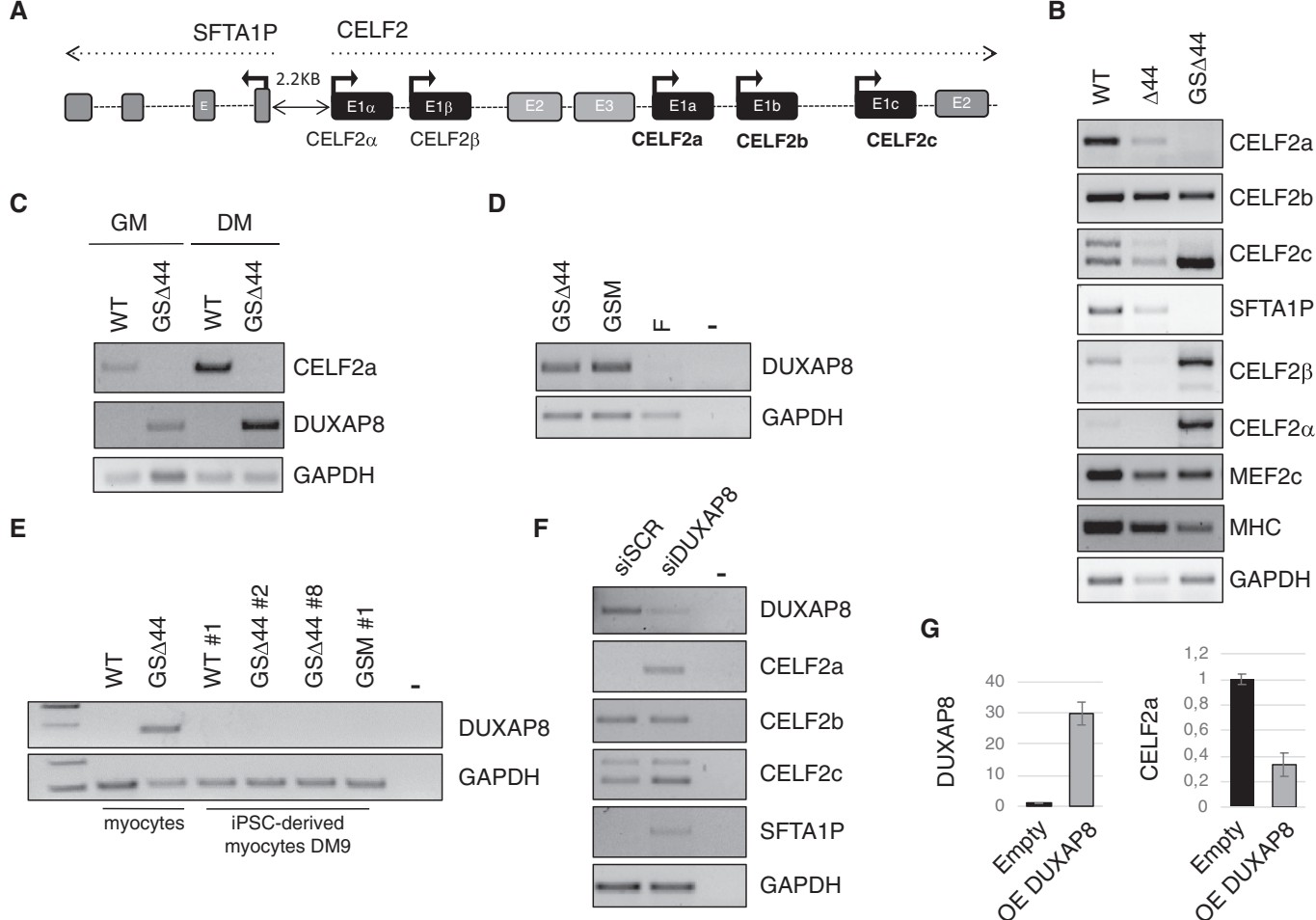

**Figure 4. A trans-acting lncRNA controls Celf2a expression.**

A Schematic representation of the exon/intron organization of Celf2 genomic locus. Arrows indicate TSS sites.

B RT–PCR analyses of indicated RNAs in control (WT), Δ44 and in GSΔ44 myocytes collected after 9 days upon the induction of differentiation. GAPDH was used as control. Representative results are shown (n = 3).

C RT–PCR analyses for CELF2a and DUXAP8 expression in WT and GSΔ44 myoblasts (GM) and myocytes (DM). GAPDH was used as control. Representative results are shown (n = 3).

D RT–PCR analysis of DUXAP8 expression in RNA extracted from GSΔ44, GSΔ44's mother (GSM) and GSΔ44's father (F) fibroblasts. GAPDH was used as control. Lane (−) indicates the negative control. Representative results are shown (n = 3).

E RT–PCR analysis for DUXAP8 expression in RNA extracted from WT and GSΔ44 myocytes and from WT (WT#1), GSΔ44 (GSΔ44#2, GSΔ44#8) and GSM (GSM#1) iPSCs induced to differentiation for 9 days. GAPDH was used as control. Lane (−) indicates the negative control. Representative results are shown (n = 3).

F RT–PCR for the indicated RNAs on extracts from GSΔ44 myoblasts treated with either control siRNAs (siSCR) or siRNAs against DUXAP8 (siDUXAP8) and collected 2 days after transfection. GAPDH was used as control. Lane (−) indicates the negative control. Representative results are shown (n = 3).

G qPCR analysis of DUXAP8 and CELF2a expression in control myoblasts transfected with a plasmid for the overexpression of DUXAP8 (OE DUXAP8) or an empty vector (Empty), and collected after 96 h. Relative mRNA levels were calculated with the delta delta Ct method. qPCRs were normalized against an internal control (GAPDH) and plotted relative to the expression level in the sample treated with the empty vector which was set to a value of 1. The mean ± SEM of triplicates from one representative experiment is shown (n = 3).

Source data are available online for this figure.

exclusive (Fig 4C). In agreement with such inverse correlation, DUXAP8 resulted to be expressed in fibroblasts of GSΔ44 and of his mother, both lacking Celf2a (Martone *et al*, 2016), and not in those of the father (Fig 4D), where Celf2a expression is unaffected (Martone *et al*, 2016). Notably, in myocytes (Myo) derived from GSΔ44 iPSCs (clones #2 and #8) and from his mother iPSCs (GSM#1), which re-express Celf2a (Fig 2A), DUXAP8 was absent (Fig 4E). Moreover, in these cells also SFTA1P expression resumed similarly to Celf2a (Fig EV4F) further demonstrating the regulatory link

between these two genes. Altogether, these data indicate a direct relationship between the DUXAP8 expression and Celf2a/SFTA1P repression.

Notably, RNAi against DUXAP8 in GSΔ44 myoblasts resumed Celf2a expression without affecting the levels of Celf2b and Celf2c (Fig 4F). Interestingly, the expression of the divergent SFTA1P transcript was also resumed upon inhibition of DUXAP8 (Fig 4F), supporting the hypothesis that this lncRNA controls both transcriptional units.

We then tested whether the exogenous expression of DUXAP8 in control myoblasts could downregulate Celf2a expression. High levels of DUXAP8 were obtained, equally partitioned among subcellular fractions (Fig EV4I). Under these conditions, we observed that in parallel to the strong DUXAP8 accumulation, a clear decrease in Celf2a expression was obtained (Fig 4G).

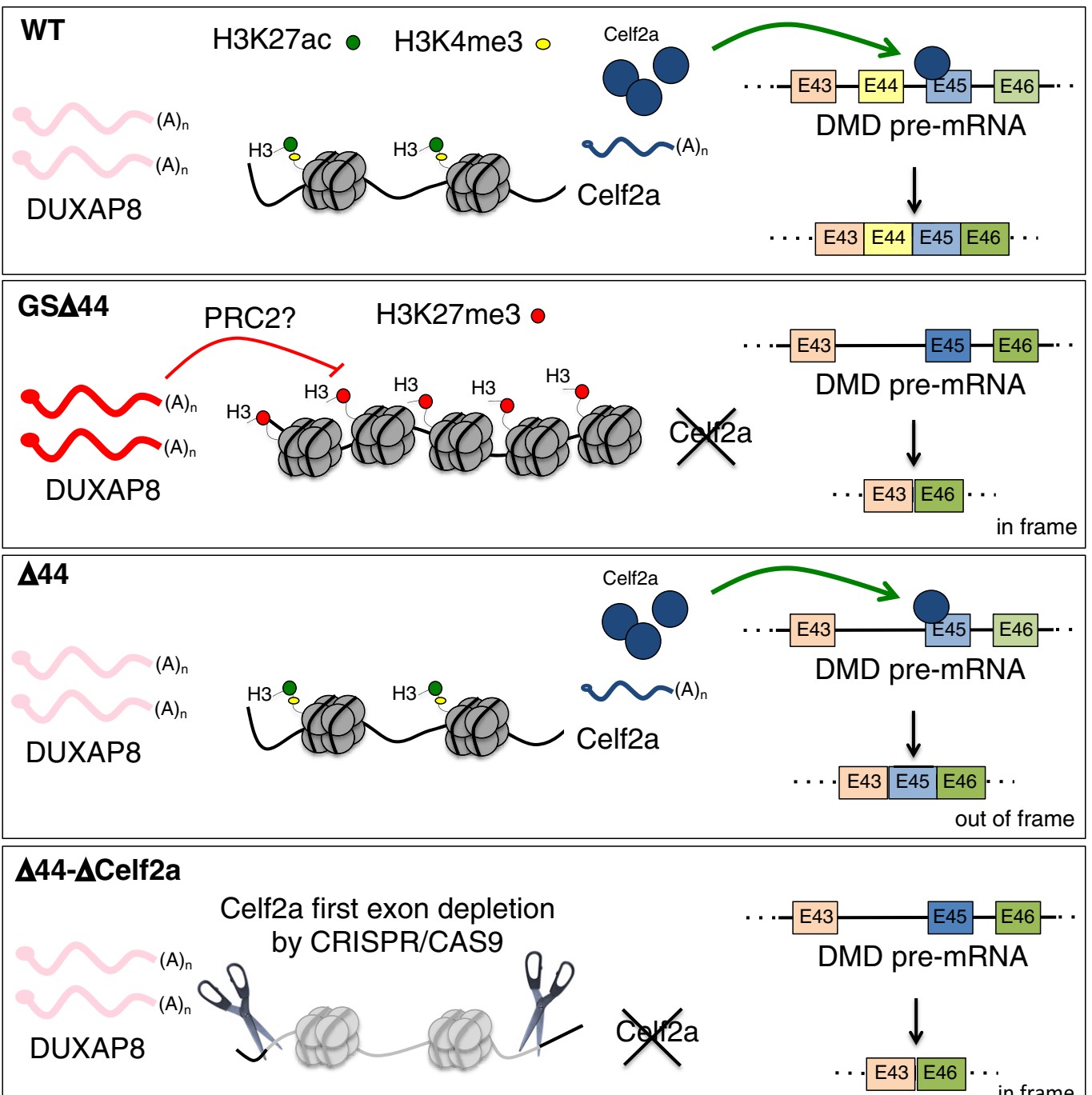

**Figure 5. Schematic representation of the four analysed situations.**

First panel: In WT condition, the lncRNA DUXAP8 is not expressed and the chromatin of Celf2a regulatory region is characterized by a permissive signature (H3K27ac and H3K4me3) that allows the expression of this factor. In the presence of Celf2a protein, the ex45 is normally included in the mature form of DMD mRNA. Second panel: In GSΔ44 cells, the lncRNA DUXAP8 is aberrantly expressed. The chromatin of Celf2a regulatory region is marked by the repressive H3K27me3, Celf2a is not expressed, and the ex45 is not included in the mature form of DMD mRNA allowing the production of an in-frame DMD transcript (Δ44–Δ45). Third panel: in Δ44 condition DUXAP8 is absent, Celf2a is expressed and ex45 is included in the mature form of DMD mRNA producing an out of frame transcript (Δ44). Fourth panel: In Δ44 cells, the depletion of Celf2a isoform is paralleled by a recovery of dystrophin protein (5%) obtained by the exclusion of ex45 from the mature DMD transcript (Δ44–Δ45).

Collectively, these results point to a functional role for DUXAP8 in the repression of Celf2a expression and suggest that increased expression of DUXAP8 in GSΔ44 (and his mother) is the primary event leading to "therapeutic" exon 45 skipping.

## Discussion

The study of the molecular mechanisms through which a mild Duchenne phenotype is obtained in the GSΔ44 case has revealed several novel peculiar features which go far beyond the direct genotype–phenotype correlation. Among them, there is the fact that the lack of Celf2a could not be attributed to any mutation in the gene, but instead to a maternally inherited epigenetic silencing of the Celf2a genomic region. Indeed, ATAC-seq and ChIP-seq for several histone marks performed in GSΔ44 myoblasts showed that the Celf2a genomic region is much less accessible than in control cells.

Notably, such repression was lost when fibroblasts from GSΔ44 were reprogrammed into iPSCs and afterwards converted into myocytes where the Celf2a isoform is specifically expressed; in such case, efficient rescue of Celf2a expression and failure of DMD exon 45 skipping were observed. Interestingly also the mother's fibroblasts, which are devoid of Celf2a expression, recovered production of such a factor when reprogrammed into iPSCs and differentiated into myocytes.

We previously showed that, differently from controls, GSΔ44 fibroblasts directly trans-differentiated into myocytes through MyoD treatment did not recover Celf2a expression (Martone et al, 2016); therefore, only reprogramming and not direct trans-differentiation allows the removal of the epigenetic repression of Celf2a. However, reprogramming has to be followed by myogenic differentiation since iPSCs are devoid of specific myogenic transcriptional activators required for Celf2a isoform transcription.

The process of reprogramming somatic cells into iPSCs has been described to reset epigenetic marks established during somatic differentiation (Maherali et al, 2007; Hewitt & Garlick, 2013). However so far, the extent of epigenetic changes associated with reprogramming and their implications in human disease is still amply debatable (Lister et al, 2011). In some cases, reprogramming enabled the reversion of pathological phenotypes linked to a very severe disease; this was observed in epigenetically compromised dermal fibroblasts, which were shown to successfully generate normal iPSCs upon reprogramming (Li et al, 2017). Here, we show an opposite case where inherited epigenetic control is clearly associated with the amelioration of a pathological state which is lost upon reprogramming.

Moreover, in this study we discovered that there is a direct correlation between the epigenetic repression of Celf2a and the expression of the lncRNA DUXAP8: in fact, this non-coding RNA is expressed in GSΔ44 fibroblasts and mature myocytes when Celf2a is repressed, while it is absent in myocytes derived from reprogrammed iPSCs when Celf2a is reactivated. In these conditions, Celf2a was re-expressed and exon 45 skipping was lost. Moreover, DUXAP8 modulation by knock down or overexpression further supported an indirect correlation with Celf2a transcription. Further studies will be required in order to underscore the molecular mechanisms regulating the expression of this lncRNA in GSΔ44 and its effects on Celf2a transcription. Interestingly, DUXAP8 proximity to centromeric regions, which are known to contain repeated sequences and extensive epigenetic repression marks, could be involved in this mis-regulation.

Finally, the demonstration that Celf2a depletion is sufficient to induce skipping of exon 45 and partial rescue of dystrophin synthesis in two different DMD genetic backgrounds indicates that ablation or inactivation of this factor could become an additional therapeutic approach for all those cases where the skipping of exon 45 can recover a functional ORF (Fig 5).

## Materials and Methods

### Generation and maintenance of human iPSCs

GSΔ44, GSΔ44's mother (GSM), GSΔ44's father (F) and UPΔ44 skin biopsies were previously obtained as already described in refs. Lenzi et al (2016), Martone et al (2016). The whole study was performed in accordance with the Declaration of Helsinki, and all the patients provided written informed consent before the inclusion in the study. Dermal fibroblasts were isolated from these explants and cultured in DMEM high-glucose supplemented with 15% FBS, 1× L-Glu and 1× penicillin–streptomycin (all from Sigma-Aldrich). For reprogramming experiments, fibroblasts in a 35-mm dish were infected in serum-free conditions and in the presence of 5 mg/ml polybrene with the lentiviral vector hSTEMCCA (Somers et al, 2010), carrying the four reprogramming factors OCT4, SOX2, KLF4 and cMYC in a single polycistronic unit. The iPSC lines were generated according to the method described in Lenzi et al (2015). Generation and characterization of WT#1 iPSCs are described in Lenzi et al (2015). All iPSC lines were maintained in NutriStem hPSC XF (Biological Industries) in plates coated with hESC-qualified Matrigel (Corning) and passaged every 4–5 days with 1 mg/ml Dispase (Gibco).

### Karyotype analysis

For cytogenetic analysis, iPSCs were treated with colchicine (10 μg/ml) overnight at 37°C, followed by an incubation of 20 min with KCl 0.56%. Then, iPSCs were fixed in methanol and acetic acid (3:1), treated with Earle's basic salt solution (BSS) at 87°C and stained with 2% Giemsa at pH 6.8 (RHG banding, R-bands by heating using Giemsa). The analysis was performed using the ECLIPSE 1000 NIKON and the GENIKON System v 3.9.8.

### Differentiation of iPSCs into muscle cells

iPSCs were passaged (passage number around 10–20) in 35-mm dishes. Differentiation into myocytes was performed as described in ref. Lenzi et al (2016) by dissociating iPSC colonies into single cells and using the co-expression of MyoD (epB-Puro-TT-mMyoD plasmid) and Baf60c (epB-Bsd-TT-Baf60c) to induce myogenic conversion (Albini et al, 2013). For the differentiation of CRISPR/Cas9-edited iPSC clones, the puromycin selection cassette present in the epB-Puro-TT-mMyoD plasmid was substituted with neomycin resistance (epB-Neo-TT-mMyoD) using In-Fusion® HD Cloning Kit (Takara Bio USA, Inc). Oligonucleotides are listed below:

> NeoR + PT (inverse) F: CCGCATGTTAGAAGACTTC
> NeoR + PT (Inverse) R: TTCCCATCTATAACAAGAAAAT
> NeoR + PT (Insert) F: TGTTATAGATGGGAAACCATGATTACGC-CAAGCTC
> NeoR + PT (Insert) R: TCTTCTAACATGCGGCGAATTAATTCTGTGGAATGTGTG

## Microscopy and image analysis

Differentiated iPSCs were fixed in 4% paraformaldehyde (Electron Microscopy Sciences, Hatfield, PA) in PBS at 4°C for 20 min. MHC immunofluorescence was performed as previously described (Cazzella *et al*, 2012), using anti-MHC (MF20 clone hybridoma supernatants) incubation O/N at 4°C. After serial washes in 1% fetal bovine serum/PBS, samples were incubated in 1% FBS/PBS with goat anti-mouse Cy5 conjugated (1:400; Jackson ImmunoResearch). The incubations were performed for 1 h at room temperature. The nuclei were stained with DAPI (4′,6-diamidino-2-phenylindole). Samples were imaged on inverted microscope Zeiss AxioObserver A1 equipped with Axiocam MRM R camera and Plan-Neofluar EC 10×/0.3 M27 objective. Images were acquired with AxioVision Rel.4.8 software. At least 7 randomly chosen microscope fields of two independent biological samples were analysed. The fusion index indicates the percentage of nuclei in multinucleated myotubes (defined by at least two nuclei) divided by total number of nuclei in a microscope field.

## RNA extraction and RT–PCR

RNA extraction was performed using Direct-zol Miniprep RNA Purification Kit (Zymo Research) with on-column DNase treatment, following the manufacturer's instructions.

Reverse transcription was performed on 300 ng of total RNA using PrimeScriptTM RT Master Mix (Perfect Real Time) (Takara), and PCR was performed using MyTaq HS DNA Polymerase (Bioline). Oligonucleotides are listed below:

> DMD E42F GAAGACATGCCTTTGGAAATTTCT
> DMD E43F CTACAACAAAGCTCAGGTCG
> DMD E46R CTCTTTTCCAGGTTCAAGTGG
> DMD E52R GATCCGTAATGATTGTTCTA
> CELF2 E1aF CTATGAGAAATGAAGAGCTGC
> CELF2 E1bF GATTTCCTCCCGGACATGACG
> CELF2 E1cF CTCTGCTCGACAGCAGCACG
> CELF2 α F GGGATAACACAGCCCAGGAAG
> CELF2 β F TCCTTGGTCTCTGACTTGGA
> CELF2 E2R GAGGACGTTGATCTGGTAGAC
> SFTA1p E1F TGTGGCACGAGTAAGCCAAA
> SFTA1p E3R GGTGGTCTGCCATCTCACTT
> LNC00710 F TCTTGGCCTCTCAGTAGACT
> LNC00710 R TTCATGCTACCTTCTATTCC
> RP1-251M9.2 F ACATTCTCCTTCCTTTCTGG
> RP1-251M9.2 R ATTTGGAATGAAGAGGTGAGT
> RP1-251M9.3 F AGAGAAGTGAGGCATGGGAT
> RP1-251M9.3 R TCATGTCTTGCAGTGCCAAT
> CELF2-AS2 F GAAGGGGAAAACGACGGACT
> CELF2-AS2 R TTGTGAGGCTGGAGTGTGAC
> CELF2-AS1 F CAGGGCACTTGATCTGGCAT

> CELF2-AS1 R CTGTAATCCACAGCACCCGT
> DUXAP8 E6 F AGGATGGAGTCTCGCTGTATTGC
> DUXAP8 E7 R GGAGGTTTGTTTTCTTCTTTTTT
> OCT4 EXOG F TCTGGGCTCTCCCATGCATTCAAA
> OCT4 EXOG R CTGACAGCCATTGGACCTGGATTT
> GAPDH F CCTTCTCCATGGTGGTGAAGAC
> GAPDH R CACCATCTTCCAGGAGTGAG

## qPCR

qPCR was performed on cDNA using PowerUp SYBR Green Master Mix (Thermo Fisher Scientific). Oligonucleotides are listed below:

> NANOG F CCAAATTCTCCTGCCAGTGAC
> NANOG R CACGTGGTTTCCAAACAAGAAA
> OCT4 F ATGCATTCAAACTGAGGTGCCTGC
> OCT4 R AACTTCACCTTCCCTCCAACCAGT
> SOX2 F TCAGGAGTTGTCAAGGCAGAGAA
> SOX2 R GCCGCCGCCGATGATTGTTATTA
> REX1 F AAAGCATCTCCTCATTCATGGT
> REX1 R TGGGCTTTCAGGTTATTTGACT
> NR2F2 F GCCATAGTCCTGTTCACCTCA
> NR2F2 R AATCTCGTCGGCTGGTTG
> DUXAP8 E7 F AAAAAAGAAGAAAACAAACCTCC
> DUXAP8 E8 R GTGCCATTTTTCTTGTGGAAACC
> MYH1: Hs_MYH1_2_SG QuantiTect Primer Assay, Cat.N. QT01671005
> Myogenin: Hs_MYOG1_SG QuantiTect Primer Assay, Cat.N. QT00001722
> Dystrophin: Hs_DMD_1_SG QuantiTect Primer Assay, Cat.N. QT00085778
> MYOD: Hs_MYOD1_1_SG QuantiTect Primer Assay, Cat.N. QT00209713
> MEF2C: Hs_MEF2C_1_SG QuantiTect Primer Assay, Cat.N. QT00053368

## Western blot analysis

Protein extracts were loaded onto a NuPAGE Tris-Acetate Minigel 3–8% 1 mm (Invitrogen). Running and blotting were performed in an XCell SureLock Minicell (Invitrogen) according to the manufacturer's instructions, and proteins were transferred to a nitrocellulose transfer membrane (Amersham Protran). Membranes were blocked with 10% non-fat dry milk and incubated O/N with primary antibodies. Protein detection was carried out with Clarity ECL Western Blotting Substrate (Bio-Rad). Primary antibodies were as follows: anti-dystrophin (NCL-DYS1 Novocastra Laboratories, 1:80); anti-actinin (ACTN sc-15335 Santa Cruz, 1:1,000) Secondary antibody was as follows: ImmunoPure Goat Anti-Mouse IgG Peroxidase Conjugated (31430 Pierce, 1:10,000).

## Genome editing using CRISPR/Cas9

sgRNA sequences were designed with a CRISPR design tool (Li *et al*, 2016) (http://crispr.mit.edu/ site, no longer available) to minimize potential off-target editing. Three top ranking guides for the region containing the first exon of Celf2a (Chr10:11047177-11047403,

GRCh37/hg19 assembly) were selected and cloned into the CRISPR/ Cas9 plasmid pX330 (Li *et al*, 2016; Addgene#42230).

The sgRNA sequences are listed below:

> sgRNA1: CACCGCTCATAGTAACAGCGGATTT
> sgRNA2: CACCGGCGTTAGTGAGCAGCGACTG
> sgRNA3: CACCGGTTTGAGCATACTTCTGAAC

The homology-directed repair construct (donor plasmid) contains 800 bp of homologous sequences flanking a floxed puromycin resistance gene (HR110 Pa-1, System Bioscience). The homologous sequences were designed to produce a 245-bp depletion in the CELF2 locus that encompass the Celf2a first exon (196-bp upstream ATG and 53-bp downstream ATG).

Oligonucleotides are listed below:

> 5′arm F: AAAAgagctcGAGCGCATTCTTGGCTTCAG
> 5′arm R: AAAAggtaccCTCCTGCAGATTCACTAAGTAGAAATAA
> 3′arm F: AACCTAGATCGGATCCGGTATGTTGTTGTGTCCTTTGT
> 3′arm R: CAGTCGACGGGGATCCCCAACCAACATAGCGGCAGA

iPSCs were co-transfected with 2 μg of donor plasmid and 1 μg of each sgRNA/CRISPR/Cas9 construct, using the Neon Transfection System (Life Technologies) as previously described (Lenzi *et al*, 2015). Selection in 0.5 μg/ml puromycin allowed to isolate only the colonies that had integrated the selection cassette, in which substitution of CELF2a first exon was confirmed by gDNA PCR with oligonucleotides listed below:

> CELF2 E1a F (F2): CTATGAGAAATGAAGAGCTGC
> 3′arm R (R2): GGTGCCTCCTTCCTGGTATG
> CASS HR F (F1): CCGGTGCCTGAAATCAACCT
> Post 3′ R (R1): GACCGAAATTGTGAGGGCAA

## Genomic DNA extraction and analysis

Genomic DNA was extracted using genomic DNA mini kit (Blood/ cultured cells) (Geneaid), following manufacturer's instructions. Oligonucleotides are listed below:

> DMD E43 F: CTACAACAAAGCTCAGGTCG
> DMD E43 R: TTGGAAATCAAGCTGGGAGAGA
> DMD E44 F: GATTTGACAGATCTGTTGAGA
> DMD E44 R: AGCATGTTCCCAATTCTCAGG
> DMD E45 F: ATGGCATTGGGCAGCGGCAAAC
> DMD E45 R: ACCTCCTGCCACCGCAGATTCA
> DMD E10 F: GCAGTTCATTGATGGAGAGTG
> DMD E10 R: CACCACTTCCACATCATTAGA
> GSΔ44 DMD deletion F: CCAGTTGATTCTTATGTGCAAC
> GSΔ44 DMD deletion R: GTGCTTCTGCGTGTGTTTG

## ATAC Sequencing and bioinformatics analyses

ATAC-seq libraries were prepared starting from 25,000 cells as in ref. Buenrostro *et al* (2015) with minor modifications. Two pair-end libraries were prepared and sequenced for each sample (50 bp).

The ATAC-seq samples were mapped against the hg19 human genome assembly using Bowtie with the option –m 1 to discard those reads that could not be uniquely mapped to one region only, and with the option –X 2,000 to define the maximum insert size for paired-end alignment (Langmead *et al*, 2009). Mitochondrial reads were removed from each resulting map, and down-sampling was applied to obtain the same number of mapped fragments per sample. Correlation between biological replicates was assessed to ensure high reproducibility before pooling each set of replicates.

Model-based analysis of ChIP-Seq (MACS) was run with the default parameters but with the shift-size adjusted to 100 bp to perform peak calling (Zhang *et al*, 2008).

The peaks reported in both conditions were gathered on a single set of peaks, and the maximum number of normalized reads within each peak in WT and GSΔ44 conditions was determined. We defined as differential peaks the ones with (i) at least twofold enrichment in one of the conditions and (ii) peaks that have normalized value > 2 in at least one of the two conditions.

The genome distribution of each set of peaks was calculated by counting the number of peaks fitted to each region's class according to RefSeq annotations (O'Leary *et al*, 2016). Promoter is the region between 2.5 kb upstream and 2.5 kb downstream of the transcription start site (TSS). Genic regions correspond to the rest of the gene (the part that is not classified as promoter), and the rest of the genome is considered to be intergenic. Peaks that overlapped with more than one genomic feature were proportionally counted the same number of times.

Each point on the scatterplots of ATAC-seq intensities between two conditions was calculated by determining the maximum value of the sample inside each peak at each condition.

## ChIP assay

ChIP assay for MyoD was performed with the MAGnify ChIP (Thermo fisher scientific) kit, according to manufacturer's instructions. 3 million WT and GSΔ44 human myoblasts were cultured in growth conditions and fixed, after 72 h, in 1% paraformaldehyde. Cells were then washed and collected in PBS. Cell extracts were sonicated using a Bioruptor Ultrasonicator (Diagenode). 6 μg of anti-MyoD (M-318: sc-760 Santa Cruz Biotechnology) or appropriate IgG control (provided in the kit) was used for immunoprecipitation overnight. The oligonucleotides used in the qPCR analyses are listed below:

> Peak A F: CTGCCAAGATCCGTCTCAG
> Peak A R: ACACAGTTGCAGGCGTAACA
> MAFA F: TTGCTTATCCCCATGGCAACT
> MAFA R: GGCCCGTCCCTACCTCTT
> LZTS2 F: GTGTCTGCATCCCTGAAGGT
> LZTS2 R: GAGTGGAGGTGAGGGTAGGA

The qPCR signal derived from the sequence of interest (calculated as percentage of INPUT) was divided by the signal derived from the positive sequence controls (calculated as percentage of INPUT). Positive controls (MAFA and LSTZ2) were selected based on the intensity of the MyoD peak that is comparable with the one observed for the peak A on Celf2 (Peak A, Fig EV3D). Moreover, the chromatin signature derived from ChIP-seq data at the control sequences did not differ between samples.

## ChIP Sequencing and bioinformatics analyses

ChIP-seq libraries were prepared starting with 20 million cells as in ref. Beringer *et al* (2016). Single-read libraries were prepared and sequenced for each sample (50 bp).

ChIP-seq samples normalized by spike-in were mapped against a synthetic genome constituted by the human and the fruit fly chromosomes (hg19 + dm3) using Bowtie with the option -m 1 to discard reads that did not map uniquely to one region only (Orlando *et al*, 2014).

Model-based analysis of ChIP-Seq was run with the default parameters but with the shift-size adjusted to 100 bp to perform the peak calling against the corresponding H3 control sample (Zhang *et al*, 2008).

The genome distribution of each set of peaks was calculated as for ATAC-seq analysis.

The aggregated plots of ChIP-seq samples containing a spike-in were generated by counting the number of reads mapped in human around the TSS for each gene and then averaging these values for the total number of reads mapped on the fruit fly genome and the number of targets of the gene list, as previously described (Taruttis *et al*, 2017).

The UCSC genome browser was used to generate the screenshots for each group of experiments along the manuscript (Kent *et al*, 2002).

## RNA Sequencing and bioinformatic analyses

Trimmomatic software (Bolger *et al*, 2014) was used in order to remove adapter sequences and poor quality bases from raw reads, setting the minimum read length after trimming to 20. Reads were mapped to human GRCh38 genome and Ensembl 90 transcriptome (Zerbino *et al*, 2018) using STAR software (Dobin *et al*, 2013), with parameters

*–outFilterType* BySJout *–outFilterMultimapNmax* 20 *–alignSJoverhangMin* 8
*–alignSJDBoverhangMin* 1 *–outFilterMismatchNmax* 999 *–outFilterMismatchNoverLmax* 0.04
*–outFilterIntronMotifs* RemoveNoncanonical.

Cuffdiff 2 software (Trapnell *et al*, 2013) was used to perform differential expression analysis using bias correction and rescue mode for multi-mapped reads, and setting the library type to fr-first-strand; reads aligning to small non-coding RNAs were ignored by providing a mask GTF file.

## Overexpressing constructs and plasmid transfection

The construct for the overexpression of CELF2a was obtained as described in ref. Martone *et al* (2016).

The construct for the overexpression of SFTA1p was obtained by cloning SFTA1p sequence in pCDNA3.1(−) plasmid (Invitrogen). The SFTA1p sequence was amplified by PCR from WT myoblast cDNA using the oligonucleotides: 5′-aaaaaGGTACCcttggcagagagcgc cct-3′ and 5′-aaaaaAAGCTTgaaagatgaatgtaagg-3′. The uppercase bases are KpnI and HindIII restriction sites.

The construct for the overexpression of DUXAP8 was obtained by cloning DUXAP8 sequence in pCDNA3.1(+) plasmid (Invitrogen).

The DUXAP8 sequence was amplified by PCR from GSΔ44 myoblast cDNA using the oligonucleotides: 5′-aaaaGGTACCgaggcccctgcagc ag-3′ and 5′-aaaaGCGGCCGCtcagaaggcttagcttgcact-3′. The uppercase bases are KpnI and NotI restriction sites.

Transient transfection of plasmids was performed using Lipofectamine-2000 (Thermo Fisher Scientific) according to manufacturer's specifications.

## RNA interference on human myoblasts

Interference with Antisense LNA GapmeR against SFTA1p was performed using Lipofectamine-2000 (Thermo Fisher Scientific) according to the manufacturer's specifications, media was replaced after 5 h with fresh media, and cells were harvested and analysed 24 h after the transfection. DUXAP8 siRNA interference was performed using Lipofectamine RNAi-MAX (Thermo Fisher Scientific) overnight, in total media without antibiotics, according to the manufacturer's specifications. The next morning, the media was replenished, and cells were harvested and analysed 48 h after the transfection. SFTA1p (GapmeR) and DUXAP8 (siRNA) and their, respectively, scrambled negative control were purchased from Qiagen.

The nucleotide sequences of siRNAs for SFTA1P and DUXAP8 are listed below:

> lnaSFTA1P CACTTGAAGAGGTGCT
> siDUXAP8 CCATACGAGAATGGGTCTAAA

The expression levels of both SFTA1p and DUXAP8 were quantified by RT–PCR. The primers are listed in the RT–PCR paragraph.

## Nucleus–cytosol fractionation

500,000 myoblasts were harvested and fractioned using the Paris Kit (Thermo Fisher scientific). GAPDH and pre-GAPDH were amplified as positive controls of fractionation.

## Chromatin–nucleus–cytosol fractionation

2 million myoblasts were harvested and fractioned as in ref. Conrad and Ørom (2017) with minor modifications.

## Muscle cell cultures and treatments

WT1, WT2, Δ44 (from the Telethon Network of Genetic Biobanks) and GSΔ44 human primary myoblasts were cultured in Human skeletal muscle growth medium (PromoCell) and grown in a humidified incubator, at 5% $CO_2$ and 37°C. Cells were induced to differentiate with human skeletal muscle differentiation medium (PromoCell; Choi *et al*, 1990). GSΔ44 primary myoblasts at early passages were co-transfected with 4 µg of epB-Puro-TT-mMyoD (or 2 µg of epB-Puro-TT-mMyoD plus 2 µg epB-Bsd-TT-Baf60c; Lenzi *et al*, 2016) and 1 µg of the piggyBac transposase using the Neon Transfection System (Life Technologies) according to the manufacturer's specifications.

## Statistical analyses

Experiments were performed in three biological replicates.

**The paper explained**

**Problem**

Duchenne Muscular Dystrophy (DMD) is a rare X-linked recessive neuromuscular disease; it is caused by mutations in the dystrophin gene that lead to the absence of the dystrophin protein. At present, possible therapeutic approaches for the cure of DMD range from exon skipping to stem cells delivery and CRISPR/Cas9 methodology. However, all these strategies are quite laborious and expensive. This is why the availability of pharmacological treatments able to provide skipping of specific exons is strongly needed. So far, only for a specific subclass of mutations including premature stop codons, a specific drug, Ataluren, is under clinical test. However, in phase III trial a significant difference between treated patients and the placebo group was not observed.

**Results**

In a previous paper, we described a case study (GSΔ44) in which the lack of a specific splicing enhancer protein (Celf2a) contributes to natural exon 45 skipping and recovery of dystrophin synthesis (Martone *et al*, 2016).

In this paper, we were able to demonstrate that Celf2a regulation occurs at the epigenetic level and is mediated by the DUXAP8 lncRNA. Moreover, the epigenetic silencing in our case study is inherited from the mother. Interestingly, upon reprogramming of GSΔ44 fibroblasts into induced pluripotent stem cells (iPSCs) the expression of Celf2a is restored, confirming the hypothesis of the epigenetic regulation. We were also able to demonstrate that Celf2a depletion, obtained by CRISPR/Cas9 strategy applied to iPSCs derived from DMDΔ44 fibroblasts, different from the GSΔ44 ones, was able to restore dystrophin synthesis upon induction of muscle differentiation.

**Impact**

We found a new therapeutic target to treat DMD pathology and identified its mechanism of regulation. Our findings pave the way to design and set up pharmacological treatments, such as available epigenetic drugs, to inhibit Celf2a expression/activity to achieve exon 45 skipping and rescue of dystrophin synthesis in a subgroup of DMD patients (almost 10%).

Statistical significance of differences between means was assessed by a two-tailed *t*-test, and $P < 0.05$ was considered significant. Normal distribution was assessed with Shapiro–Wilk test.

## Data availability

The data sets produced in this study are available in the following databases:

- ATAC-seq and ChIP-seq: National Center for Biotechnology Information Gene Expression Omnibus (Barrett *et al*, 2013) GSE139571 (https://www.ncbi.nlm.nih.gov/geo/query/acc.cgi?acc=GSE139571)

**Expanded View** for this article is available online.

## Acknowledgements

The authors acknowledge GS and his family for their active and friendly collaboration, O. Sthandier and M. Marchioni for technical help, D. Cacchiarelli for scientific advice, V. Silenzi for reading the manuscript, the Telethon Network Genetic Biobanks (GTB12001F) for biological samples and SMART by Servier (https://smart.servier.com/) for figure design. This work was partially supported by grants from ERC-2019-SyG (855923-ASTRA), Telethon (GGP16213), Parent Project Italia, AIRC (IG 2019 Id. 23053) and PRIN 2017 (2017P352Z4) to I.B.; Spanish Ministry of Economy, Industry and Competitiveness (MEIC) (BFU2016-75008-P) to L.D.C. and Sapienza Research Calls (RM118164363B1D21) to J.M.

## Author contributions

Conceptualization and design of the work were carried out by JM, LDC and IB. The experiments were performed and analysed by JM, ML, FC, AR, VDC, DM, TS and LP. Bioinformatics data analysis was performed by EB, AS and AC. The original draft of the manuscript was written by JM and IB. The draft was reviewed and edited by all the authors.

## Conflict of interest

The authors declare that they have no conflict of interest.

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
