## [Review Process File · EMBO Molecular Medicine]

Trans-generational epigenetic regulation associated with the amelioration of Duchenne Muscular Dystrophy

Julie Martone, Michela Lisi, Francesco Castagnetti, Alessandro Rosa, Valerio Di Carlo, Enrique Blanco, Adriano Setti, Davide Mariani, Alessio Colantoni, Tiziana Santini, Lucia Perone, Luciano Di Croce and Irene Bozzoni

DOI: [10.15252/emmm.202012063](https://doi.org/10.15252/emmm.202012063)

Corresponding author: Irene Bozzoni (irene.bozzoni@uniroma1.it)

Review Timeline:	Transfer from The EMBO Journal to EMBO	
	Molecular Medicine:	22nd Jan 20
	Editorial Decision:	26th Feb 20
	Authors' Correspondence:	11th Mar 20
	Editorial Correspondence:	11th Mar 20
	Editorial Correspondence:	19th Mar 20
	Revision Received:	27th Apr 20
	Editorial Decision:	28th May 20
	Revision Received:	5th Jun 20
	Accepted:	8th Jun 20

Editor: Celine Carret

Transaction Report:

26th Feb 2020

Dear Prof. Bozzoni,

Thank you for the submission of your manuscript to EMBO Molecular Medicine. We have now heard back from the three referees whom we asked to evaluate your manuscript.

You will see from the set of reports pasted below that the three referees find the study to be interesting, well performed and convincing. Still, all referees suggest additional experiments to better support and strengthen the main findings by not only providing necessary controls but also mechanistic understanding. Indeed Ref. #1 would like to see more iPSCs characterisation and appropriate controls, and ref. #2 and #3 suggest to provide more mechanistic insights and supporting data.

We would therefore welcome the submission of a revised version within three months for further consideration and would like to encourage you to address all the criticisms raised as suggested to improve conclusiveness and clarity. Please note that EMBO Molecular Medicine strongly supports a single round of revision and that, as acceptance or rejection of the manuscript will depend on another round of review, your responses should be as complete as possible.

I look forward to receiving your revised manuscript.

Yours sincerely,

Celine Carret

Celine Carret, PhD
Senior Editor
EMBO Molecular Medicine

*Additional important information regarding figures and illustrations can be found at <http://bit.ly/EMBOPressFigurePreparationGuideline>

***** Reviewer's comments *****

Referee #1 (Comments on Novelty/Model System for Author):

Please see remarks to author section.

Referee #1 (Remarks for Author):

The manuscript by Martone et al shows that lack of the splicing factor Celf2a results in exon skipping and partial dystrophin rescue. This is linked to a maternally inherited trans-generational epigenetic silencing likely mediated by expression of the repressive long non coding RNA DUXAP8. The work proposes an interesting and elegant mechanism of dystrophin regulation and a possible new therapeutic strategy. However, there are issues which needs to be addressed, namely:

- 1) Please provide morphological evidence and quantification of iPSC myogenic differentiation and fusion (i.e. myosin heavy chain staining, differentiation and fusion index) for all lines used in the study. This is critical to understand how reliable is the modeling platform.
- 2) "LMdelta44 clones were validated as bona fide iPSCs by qPCR assessment of pluripotency marker over-expression (NANOG, OCT4, SOX2, REX1)": please note that pluripotency-associated markers should not be over-expressed in iPSCs, but rather the opposite: the exogenous factors should be silenced and the endogenous expressed at normal hESC-like levels. Please provide evidence of the status of the exogenous reprogramming factors: this is critical in view of the use of a lentiviral reprogramming method such as the one used in this study (LVs are more resistant to silencing than conventional retroviruses).
- 3) "Altogether these data indicate, in line with the observation that the Celf2a gene of GS44 does not show any obvious mutation (unpublished), that the lack of Celf2a expression in GS44 is due to epigenetic silencing and that this regulation is lost upon the reprogramming process". Please show evidence of lack of mutations in GS44. Also, could the presence of Celf2a in iPSC-derived myocytes be a hallmark of their lack of maturation, making them not a suitable modeling platform to study the proposed DMD mechanism? The authors should validate this finding with an alternative iPSC differentiation protocol (ideally not transgene-based).
- 4) Importantly, the authors need to provide evidence that DUXAP8 (over)expression actually induces dystrophin protein production.
- 5) The authors hypothesize inactivation/KO of Celf2a as a therapeutic strategy. What could be the off targets effects on the overall splicing machinery resulting from such a strategy? Experimental evidence on this topic would be appreciated.
- 6) Figure 1 and 2: provide quantification for skipped bands in RT PCRs and Western blots.

7) Karyotype analysis of newly generated iPSC lines would be required to match current international standards (eg <https://hpscereg.eu>). STR analysis would also be highly recommended to fingerprint iPSCs and parental fibroblasts.

8) Abstract: "inactivation of the splicing factor Celf2a was proven to be curative in other DMD backgrounds": the word curative is misleading, particularly for patients. Also, dystrophin is not rescued here, at best it would be partially rescued. Please tone down this statement.

9) Opening introductory sentence: I would use "most" DMD mutations rather than "all".

10) Are the mutations acronyms containing patients' initials? If that is the case this needs amending as might put patient confidentiality at risk.

11) Please provide a diagram of the proposed model as final figure.

12) Title: I would argue that the mechanism shown in the manuscript is "associated with" rather than "drives" DMD amelioration.

13) The authors propose Celf2a ablation or inactivation as a novel therapeutic strategy for DMD. However, they should also discuss the feasibility and competitiveness of such a strategy with current approaches in clinical/preclinical trials. For example, it might be difficult to justify exposing (and immunizing) a DMD child to an AAV knocking out Celf2a by CRISPR and inducing only 5% dystrophin when the same AAV could induce higher dystrophin levels via permanent exon skipping or by delivering a microdystrophin.

Referee #2 (Remarks for Author):

In this manuscript Martone et al. follow up a very interesting finding they published few years ago, reporting on a DMD patient with deletion of exon 44, showing the clinical milder phenotype of BMD. They discovered that this phenotype was due to the spontaneous skipping of exon 45, leading to partially dystrophin recovery, because of the lack of the splicing factor Celf2a.

These authors now present evidence that Celf2a deficiency is due to a maternally inherited trans-generational epigenetic silencing that coincides with the expression levels of the long noncoding RNA, DUXAP8. Interestingly, the authors also show that DUXAP8 expression is lost during hiPSC reprogramming into skeletal myocytes, coinciding with the loss of exon skipping and dystrophin recovery. Experiments of CRISPR/Cas9-mediated inactivation of Celf2a showed a clear causal relationship between Celf2a expression and exon 45 skipping. Combined ATACseq and ChIPseq analysis of histone marks of promoter and enhancer elements revealed the presence of potential enhancers at Celf2a locus with decreased H3K27 signal, MyoD binding and chromatin accessibility, which suggest reduced enhancer activity, in GSDelta44 muscle cells. Finally, the re-analysis of RNAseq dataset from Martone et al. 2016 revealed that the chromatin-associated lncRNA DUXAP8 is expressed in GSDelta44 muscle cells, but not controls, and further experiments established an inverse correlation between DUXAP8 and Celf2a.

Overall, this manuscript reports on a very interesting functional relationship between one splicing factor (Celf2a), a chromatin-associated lncRNA (DUXAP8), and exon 44 skipping, leading to partial dystrophin expression in a DMD patient. This information is novel and of high clinical relevance. The experiments are well executed and correctly interpreted.

Unfortunately, the authors were not able to provide mechanistic insights into the most interesting

issues opened by the data, and the manuscript in general suffers from this lack of information. Given the overall novelty and relevance of the finding, this reviewer is in favor of a publication of this manuscript in EMBO Molecular Medicine, as long as the authors address the following points.

1) Figure 2A clearly shows that re-expression of Celf2a occurs during the process of myogenic differentiation of hiPSCs and not during the reprogramming of patient-derived fibroblasts into hiPSCs. The experiment shown in S2E indicates that MyoD and BAF60C overexpression in myocytes does not recover the expression of Celf2a. While this result formally excludes that Celf2a re-expression in GSdelta44 ipSC-derived myocytes is due to an artificial effect of the ectopic expression of MyoD and BAF60C in hiPSCs, it does not address the most important issue here - whether and Celf2a expression is a feature of hiPSC differentiation into somatic differentiated cells. While this reviewer understands that this is not a central aspect of the manuscript, the inclusion of this intriguing data in the manuscript prompts a number of questions of general cell biology and mechanistic insights. Not meaning to ask the authors to identify the mechanism responsible for this phenomenon, I still believe that it is important to define whether Celf2a re-expression is generally associated with hiPSC differentiation into specialized cell types. To this purpose, I recommend that the author just validate Celf2a re-expression upon GSdelta44 ipSC differentiation into skeletal muscle, by using an alternative protocol with differentiation cues (such as those recently established in A. Pyle, O. Purquie' or G. Lee labs), and evaluate whether Celf2a re-expression occurs when GSdelta44 ipSC are induced to differentiate into at least one alternative lineages - e.g. neuronal, cardiomyocytes, hematopoietic, etc).

2) Combined ATACseq and ChIPseq analysis revealed the presence of potential regulatory elements at Celf2a locus; however, it is not clear whether they are promoters or enhancers, and whether they actually loop to regulate Celf2a promoter. Moreover, it is unclear whether changes in H3K27 and H3K4me3 signal, MyoD binding and chromatin accessibility, eventually influence these interactions. As it stands, this analysis remains quite inclusive. There are currently HiC maps available on high order chromatin interactions instigated by MyoD during activation of skeletal myogenesis in humans cells (Dall'Agnese et al. 2019) that could help capturing interactions between the regulatory elements identified in the Celf2a locus. The authors should use this resource and then exploit 3C analysis to experimentally monitor potential differences in these interactions between GSdelta44 myocytes and controls

3) I note that one major differential ATACseq peak between GSdelta44 myocytes and controls locates within the Celf2c coding gene sequence and escaped the attention of the authors. This peak also coincides with MyoD peak and H3K27ac signal in WT myocytes that is lost in GSdelta44 myocytes. I am wondering why the authors have not considered this as potential regulatory element.

4) The relationship between expression of Celf2a and DUXAP8 is remain quite correlative in the absence of mechanistic evidence. Whether and how DUXAP8 regulates Celf2a by direct chromatin interaction at regulatory sequences (e.g. promoter, enhancers) is not addressed. I note from Fig. 3A that in GSdelta44 myocytes loss of H3K27Ac and H3K4me3, as well as reduction of chromatin accessibility, coincide with increased H3K27me3 signal, which reflects presence of active PRC2 complex. PRC2-mediated histone methylation has been associated with recruitment of regulatory lncRNA. EzH2 inhibition by soluble molecules or by siRNA-mediated downregulation should be used in conditions of DUXAP8 to determine whether it can counter the repressive effect on Celf2a. This evidence would at least indicate a potential mechanism of Celf2a repression by DUXAP8, via recruitment to a PRC2-bound regulatory element.

Referee #3 (Comments on Novelty/Model System for Author):

The authors used both primary as well as hiPSC from patients and controls.

Referee #3 (Remarks for Author):

Martone et al continue their characterization of an unusual DMD patient (GS Δ 44) displaying a mild phenotype despite deletion of Dystrophin exon 44. In a previous report, they showed that GS Δ 44 lack expression the splicing factor Celf2a allowing for skipping of Dystrophin exon 45 and partial rescue of Dystrophin protein expression. In the current manuscript, the authors investigate the mechanism responsible for Celf2a silencing in GS Δ 44. Intriguingly, they found that Celf2a repression is lost when GS Δ 44 fibroblasts are reprogrammed into iPSCs. This suggests that Celf2a is repressed by an epigenetic silencing mechanism, which was supported by ChIP-seq and ATCA-seq data. Next, the authors identified a lncRNA (DUXAP8) whose expression is inversely correlated to that of Celf2a and provide data in line with the possibility that DUXAP8 represses Celf2a. All in all, the study offers a very intriguing molecular perspective on the relevance of intergenerational epigenetic disease modifiers. Nevertheless, there are a number of aspects that need to be addressed to more firmly support the authors conclusions, as listed below.

1. In Figure S2C, it is unclear what the GS Δ 44 DMD deletion means. My understanding is that the deletion of exon 44 is present only in GS Δ 44 and not in GSM. What is the band present in GS Δ 44 and GSM, but not in WT cells, in the GS Δ 44 DMD deletion lane?

2. What is the H3K4me3, H3K27Ac and MyoD signal at the Celf2a genomic region upon reprogramming (iPSC) and differentiation (iPSC-derived myocytes)? Is there a rescue of H3K4me3, H3K27Ac, chromatin accessibility and MyoD enrichment at the Celf2a genomic region upon reprogramming and/or myogenic differentiation of iPSCs?

3. DUXAP8 is highly homologous to the protein-coding gene DUXA and to other DUXA-related transcripts. It is therefore essential that the authors demonstrate that their reagents are DUXAP8-specific. For example, are the primers used to detect DUXAP8 selective for this transcript? Moreover, expression of DUXA and the other DUXA-related pseudogenes must be shown in all instances in which DUXAP8 is manipulated to demonstrate DUXAP8 siRNAs and OE are not affecting the expression of DUXA and the other DUXA-related pseudogenes.

Which is the physiological expression pattern of DUXAP8? In which human tissue is DUXAP8 normally expressed? Is the expression of DUXAP8 and Celf2a physiologically also mutually exclusive? In other words, is DUXAP8 expressed only in pathological conditions or is DUXAP8 involved in the regulation of Celf2a expression in normal physiology?

4. There are a number of mechanistic aspects that must be addressed.

What is the mechanism responsible for the differential DUXAP8 expression between the GSM, GSF and GS Δ 44 and between the primary fibroblasts and the iPSCs and iPSC-derived myocytes? Are the ChIP-seq and ATAC-seq data available to the authors showing any significant change at the DUXAP8 locus?

What is the mechanism by which DUXAP8 regulate Celf2a expression? Is it direct? Is DUXAP8 associated to the Celf2a genomic region? DUXAP8 has been reported to regulate gene expression by positively regulating PRC2 and LSD1, and by sponging miR-422a and miR-577. Are PRC2 and/or LSD1 differentially enriched at the Celf2a locus in WT vs GS Δ 44, siSCR vs siDUXAP8, and in Empty vs OE DUXAP8 cells? Is Celf2a a miR-422a and/or miR-577 target?

Dear Dr Carret,

I am writing with regard to the revision of manuscript *EMM-2020-12063-T*

The reviewers have raised several comments which we are going to address with all the required attention. In reading them I found that some of them, while being quite onerous in terms of man work and services to be used, may only add side information to the main message of the paper. As you know in Italy due to the restrictions imposed by the Covid-19 outbreak we have quite many restrictions in accessibility to the lab and safety issues for the people working there. However, we properly managed to organize ourselves, though with much longer times than expected and with great difficulties. Working with iPSC does not simplify things. Therefore, I am in the condition to prioritize our experiments; this is why I am asking your help in identifying whether there are issues that are not strictly necessary to ensure the quality of the work and the solidity of the conclusions.

I am listing here below some of the points on which I would like to ask your opinion:

- provide morphological evidence and quantification of iPSC myogenic differentiation (Ref#1, point 1)
For all the lines we show that myogenin and Mef2C, common markers of myogenic differentiation, are indeed activated during iPSC to myoblasts differentiation (see ext data Fig.1 and 2). Note that in wt conditions, and in D44 cells knocked out for Celf2a, the late differentiation marker dystrophin is produced (Fig. 1) indicating again the occurrence of appropriate myogenic differentiation.

Produce the data requested would mean to re-amplify all the iPSC lines; these data would probably not add very much to those already presented.

- Karyotype analysis of newly generated iPSC lines would be required to match current international standards (Ref#1, point 7)

We do not have the expertise ourselves to perform karyotypic analyses; moreover, at present we can't get any help from our close colleagues. We found a company in the North of Italy but also they are quite inaccessible for the time being. Besides these intrinsic difficulties, I think that this specific concern is not so strong in our case for the following reasons: as far as the issue that the lack of Celf2a is per se sufficient to recover dystrophin synthesis, it is important to note that we are comparing KO cell lines with their corresponding iPSC. Therefore, it is unlikely that CRISPR could alter the karyotype and in more than one clone in a similar manner. For the effect of reprogramming on Celf2a expression, we are also confident that the observed effect is not due to chromosomal alterations since we get the same effect on multiple independent clones both from GSD44 and from his mother.

- I recommend that the author just validate Celf2a re-expression upon GSdelta44 iPSC differentiation into skeletal muscle, by using an alternative protocol with differentiation cues (Ref#2, point 1)

It is not clear why we should use an unrelated differentiation protocol since the method utilized has been already published and here confirmed to lead to correct myogenic marker expression including dystrophin. A further prove that differentiation per se does not reactivate Celf2a comes from different evidences: if you trans-differentiate fibroblasts or myoblasts of GSD44 into myotubes, Celf2a is never reactivated. Moreover, SFTA1P, which is an unrelated transcript originating from the same genomic region of Celf2a and aberrantly silenced in GSD44, is re-activated in a manner similar to Celf2a upon reprogramming into iPSCs. Notably, SFTA1P is expressed already in iPSCs therefore, it is a good control in order to prove that the epigenetic modification of the entire region has occurred during reprogramming, while the expression of Celf2a requires progression into myogenic differentiation.

- What is the H3K4me3, H3K27Ac and MyoD signal at the Celf2a genomic region upon reprogramming (iPSC) and differentiation (iPSC-derived myocytes)? Is there a rescue of H3K4me3, H3K27Ac, chromatin accessibility and MyoD enrichment at the Celf2a genomic region upon reprogramming and/or myogenic differentiation of iPSCs?(Ref#3, point 2).

This is quite a challenging request since with iPSCs it is difficult to have enough material to perform all these analyses. Moreover it is a very long experiment and with the ongoing restrictions it will take even longer. But other than that I think that we have sufficiently shown that the modulation occurs at the level of transcription and not at other levels (stability, translation or other).

I would greatly appreciate your opinion and suggestions.

My best regards

Irene Bozzoni

Dear Prof. Bozzoni,

Thank you for your e-mail describing the difficulties you are currently experiencing. We fully understand the stressful and frustrating situation in Italy and in all COVID-19 heavily affected areas. It's unfortunately very likely that very soon we will all face similar conditions.

Please rest assured that we are happy to extend the deadline for revision and you should take the time you need to perform the experiments needed to improve the current article as suggested by the referees.

I would like to add that if you want to argue against performing an experiment that you don't think is necessary, it's perfectly fine but should be done independently of the COVID-19 special circumstances.

Most of the issues you have listed are dependent on the iPSC cells, which require time, and the suggested experiments are meant as controls of the quality and ID of these cells: the molecular markers, the different differentiation protocol, even the karyotype analysis. As such, we feel that these are important experiments to perform. The epigenetic analysis sounds like a lengthy experiment to do, but as I said, it's absolutely fine to take the time you need.

I'm happy to inquire with an external expert on iPSC and ask whether all the quality control experiments requested are really necessary if you want. Still, in the meantime, please proceed with revision as is. I will get back to you as soon as I hear back from this expert.

With my best wishes,

Celine

Celine Carret, PhD
Senior Editor
EMBO Molecular Medicine
EMBO Press
celine.carret@embo.org
www.embopress.org/journal/17574684
tel. +49(0) 6221 8891 411

Follow us on Twitter @EmboMolMed
Sign up for eTOCs at embopress.org/alertsfeeds

Dear Irene,

Thank you for your note and sorry for not having replied earlier. As a matter of fact I only just received the advice I sought regarding the points you made. The adviser was not immediately available, I guess the COVID-19 situation has long reaching repercussions.

To answer to your specific comments, please see the answers as follows:

- provide morphological evidence and quantification of iPSC myogenic differentiation (Ref#1, point 1)

For all the lines we show that myogenin and Mef2C, common markers of myogenic differentiation, are indeed activated during iPSC to myoblasts differentiation (see ext data Fig.1 and 2).

> Our advisor said the following: "some immunofluorescence images showing typical morphology of myocytes, with some typical markers expressed at high levels in the right subcellular locations, such as the ones in the original publication describing the protocol (<https://www.ncbi.nlm.nih.gov/pmc/articles/PMC5009183/>) [would be needed]. Some quantification to get an idea about homogeneity would be a plus. Another important assay would be to show functionality, which is standard for other cell types [...]. The last two are not essential, but typically requested by higher impact journals. "

Note that in wt conditions, and in D44 cells knocked out for Celf2a, the late differentiation marker dystrophin is produced (Fig. 1) indicating again the occurrence of appropriate myogenic differentiation.

> Our advisor is unsure whether one marker assessed by western blot is accepted in the muscle differentiation field as marker for proper differentiation, it isn't in other fields. This said, should you add some immunofluorescence as recommended above, this issue would be solved.

- Karyotype analysis of newly generated iPSC lines would be required to match current international standards (Ref#1, point 7)

We do not have the expertise ourselves to perform karyotypic analyses; moreover, at present we can't get any help from our close colleagues. We found a company in the North of Italy but also them are quite inaccessible for the time being. Besides these intrinsic difficulties, I think that this specific concern is not so strong in our case for the following reasons: as far as the issue that the lack of Celf2a is per se sufficient to recover dystrophin synthesis, it is important to note that we are comparing KO cell lines with their corresponding iPSC. Therefore, it is unlikely that CRISPR could alter the karyotype and in more than one clone in a similar manner. For the effect of reprogramming on Celf2a expression, we are also confident that the observed effect is not due to chromosomal alterations since we get the same effect on multiple independent clones both from GSD44 and from his mother.

> our advisor agrees "that there is a low likelihood that the specific phenotypes are caused by chromosomal aberrations, rather than the studied alterations. However, karyotyping is clearly an accepted and required standard quality control that is asked for to make sure the line does not have other flaws that could affect downstream analyses. Once published, this line may become a resource for follow-up studies and these will require a certain standard. It is also accepted in the field that some assay for fingerprinting should be done. My suggestion would be to ask for molecular karyotyping which uses a genome-wide SNP profile for analysis and also provides fingerprinting info. This only requires some genomic DNA, not live cells, which can be sent to other countries, which are still operational. We use Life & Brain in Bonn for this service."

- I recommend that the author just validate Celf2a re-expression upon GSdelta44 ipSC differentiation into skeletal muscle, by using an alternative protocol with differentiation cues (Ref#2, point 1)

It is not clear why we should use an unrelated differentiation protocol since the method utilized has been already published and here confirmed to lead to correct myogenic marker expression including dystrophin. A further prove that differentiation per se does not reactivate Celf2a comes from different evidences: if you trans-differentiate fibroblasts or myoblasts of GSD44 into myotubes, Celf2a is never reactivated. Moreover, SFTA1P, which is an unrelated transcript originating from the same genomic region of Celf2a and aberrantly silenced in GSD44, is re-activated in a manner similar to Celf2a upon reprogramming into iPSCs. Notably, SFTA1P is expressed already in iPSCs therefore, it is a good control in order to prove that the epigenetic modification of the entire region has occurred during reprogramming, while the expression of Celf2a requires progression into myogenic differentiation.

> please make sure to argue your point convincingly in writing. No experiments needed.

- What is the H3K4me3, H3K27Ac and MyoD signal at the Celf2a genomic region upon reprogramming (iPSC) and differentiation (iPSC-derived myocytes)? Is there a rescue of H3K4me3, H3K27Ac, chromatin accessibility and MyoD enrichment at the Celf2a genomic region upon reprogramming and/or myogenic differentiation of iPSCs?(Ref#3, point 2).

This is quite a challenging request since with iPSCs it is difficult to have enough material to perform all these analyses. Moreover it is a very long experiment and with the ongoing restrictions it will take even longer. But other than that I think that we have sufficiently shown that the modulation occurs at the level of transcription and not at other levels (stability, translation or other).

> please make sure to argue your point convincingly and tone down your conclusions accordingly.

As I mentioned before, we are happy to extend the revision deadline for how long as you will need it, without impacting on the scooping protection. I realise that this answer may not have been exactly what you expected but I hope that you will still find it acceptable.

Best regards,
Celine

Celine Carret, PhD
Senior Editor
EMBO Molecular Medicine

Detailed replies to the reviewers:

Ref#1

1) Please provide morphological evidence and quantification of iPSC myogenic differentiation and fusion (i.e. myosin heavy chain staining, differentiation and fusion index) for all lines used in the study. This is critical to understand how reliable is the modeling platform.

We have performed such analyses. We have added morphological evidences that iPSCs are correctly converted in myotubes by performing immunostaining analyses for myosin heavy chain (MHC, see the new Figures 1B and 2B). We also demonstrate that the fusion index (reported in the new Figures 1C and 2C) is similar for all the clones analysed with no significant differences between control and DMD cells.

These data not only demonstrate the effectiveness of our protocol in making myotubes but also that the different cell lines do not have alteration in the myogenic process.

2) "LMdelta44 clones were validated as bona fide iPSCs by qPCR assessment of pluripotency marker over-expression (NANOG, OCT4, SOX2, REX1)": please note that pluripotency-associated markers should not be over-expressed in iPSCs, but rather the opposite: the exogenous factors should be silenced and the endogenous expressed at normal hESC-like levels. Please provide evidence of the status of the exogenous reprogramming factors: this is critical in view of the use of a lentiviral reprogramming method such as the one used in this study (LVs are more resistant to silencing than conventional retroviruses).

According to the referee observation we substituted the word "over-expression" with "expression", in fact we wanted to state that pluripotency marker were over-expressed if compared to their fibroblasts counterpart, but not to normal hESC-like levels. Moreover, as suggested, we provided evidences about the silencing of one of the exogenous reprogramming factors, Oct4, adding a new panel in the Figure EV1C.

We checked only for the exogenous Oct4 factor since fibroblasts from skin biopsies were reprogrammed into iPSCs (Somers et al., 2010) using a lentiviral vector (hSTEMCCA - Somers et al., 2010, Sommer et al., 2012) carrying the four reprogramming factors OCT4, SOX2, KLF4 and cMYC in a single polycistronic unit. It is known that the transcriptional silencing of reprogramming factors is mediated by methylation of CpG contained in the LTR sequence, suggesting that the results obtained for one of them (OCT4) can be extended to the others (as already assumed in Lenzi et al., 2015).

For one iPSC clone, we were not able to observe a decrease in the exogenous OCT4 level, therefore the analysis on this clone (GSΔ44#5) was removed from the manuscript, and modified the related figure panels (Figure 2A, Figure 4E and Figure EV2B, EV2D and EV2E)

3) "Altogether these data indicate, in line with the observation that the Celf2a gene of GS44 does not show any obvious mutation (unpublished), that the lack of Celf2a expression in GS44 is due to epigenetic silencing and that this regulation is lost upon the reprogramming process". Please show evidence of lack of mutations in GS44.

Sorry for not being clear enough. In our previous paper (Martone et al., 2016) we sequenced 1.3kb DNA sequence upstream the TSS and 200nt of the first exon/intron of CELF2a and only two variants of annotated single-nucleotide polymorphisms (chr10:11046386 T/C, chr10:11047206 C/G of the hg19 human assembly) were found.

Moreover, the remaining part of the coding sequence is in common with CELF2b and CELF2c whose expression resulted to be unaffected in our patient. More importantly, upon reprogramming of the GS fibroblasts and their conversion into myoblasts, the Celf2a mRNA and protein were produced and exon 45 skipping was observed, thus demonstrating that in GS Δ 44 there are no mutations which could affect either the expression of the gene or the function of the protein.

Also, could the presence of Celf2a in iPSC-derived myocytes be a hallmark of their lack of maturation, making them not a suitable modeling platform to study the proposed DMD mechanism? The authors should validate this finding with an alternative iPSC differentiation protocol (ideally not transgene-based).

The analysis performed in response to point 1 (MHC immunofluorescence and fusion index) already indicate that the different iPSC lines are able to reach the myotube state. Moreover, we show that in WT iPSCs, Celf2a expression is associated with terminal muscle differentiation as witnessed by the co-expression of dystrophin (see Figure 1). The same situation was found for all the other iPSCs analyzed, such as the case of D44 cells KO for Celf2a where dystrophin synthesis is recovered at day 9. Note that only reprogramming is able to resume Celf2a expression in GS Δ 44; in fact, we previously demonstrated that trans-differentiation of GS Δ 44 fibroblasts into myoblasts did not recover Celf2a expression (Martone et al., 2016). Therefore is not the stage of myogenic differentiation but instead the reprogramming event that allows GS to recover Celf2a expression.

With reference to the differentiation protocol, we have previously experienced in Lenzi et al. (2016) two different procedures for converting iPSC into myotubes. In the first one only MyoD was employed and in this case terminal differentiation was reached at day 11 (dystrophin expression was used as a terminal differentiation marker). In the same paper we showed that the additional use of Baf60 allowed to shorten the differentiation process with dystrophin expression appearing at day 9. In this work, by using the MyoD/Baf60 procedure we indeed get Celf2a expression at day 9 together with dystrophin (see Figure1), while with the previous MyoD-only procedure Celf2a was visualized at day 11 (see Figure below).

Figure I. Celf2a expression is associated with terminal muscle differentiation. Celf2a is not expressed in WT iPSCs while it is turned on after differentiation to myotubes obtained by the overexpression of MyoD. It is interesting to note that Celf2a expression parallels DMD expression (lower panel). Moreover, with this protocol, while Celf2b and c are already present in iPSCs, Celf2a appears only in myotubes

4) Importantly, the authors need to provide evidence that DUXAP8 (over)expression actually induces dystrophin protein production.

Our OE conditions are such that Celf2a expression is only reduced but not completely silenced (we were anyhow lucky to manage to get some repression through exogenous expression). Such condition even if favorable to see some repression of Celf2a is not enough to get sufficient exon 45 skipping. Indeed we previously demonstrated that to achieve exon 45 skipping a strong depletion of Celf2a protein is needed (Martone et al., 2016).

5) The authors hypothesize inactivation/KO of Celf2a as a therapeutic strategy. What could be the off targets effects on the overall splicing machinery resulting from such a strategy? Experimental evidence on this topic would be appreciated.

In previous work (Martone et al., 2016) RNAseq data obtained from control, D44 and GSD44 myoblasts and myocytes were analysed. The exon skipping events possibly related to the absence of Celf2a were identified by MATS (Shen, S. et al. MATS: a Bayesian framework for flexible detection of differential alternative splicing from RNA-Seq data. *Nucleic Acids Res.* 40, e61–e61, 2012). The skipping variants related to the pathological state were excluded considering only those that were concordant in both GSD44 versus control and GSD44 versus D44-1. Only 29 coding genes were identified to be differentially spliced and, among them, only 5 were related to myogenic conditions (Figure II). Two of them, CSDE and TMED2, were further validated showing that the splicing variants observed in GSD44 maintained the ORF. From that analysis we concluded that, even if it is not possible to exclude secondary detrimental effects, no relevant modifications occur. However, despite these analyses, it is important to underline that the GSD44 study case is now 22 year old and does not show any adverse effect in any other specific physiological activities; moreover, the absence of Celf2a did not show adverse effects also in his mother.

Figure II. Alternative splicing analysis from the previous paper (Martone et al., 2016)

		Sample A		Sample B		Type	N° events
		Sample A		Sample B		(Sample A: Sample B)	
Myoblasts	GS Δ44	WT	SE	108	(63:45)		
			MXE	33	(19:14)		
			A5SS	3	(0:3)		
			A3SS	14	(9:5)		
RI	55	(17:38)					
Myotubes	GS Δ44	Δ44-1	SE	33	(10:23)		
			MXE	17	(9:8)		
			A5SS	5	(0:5)		
			A3SS	2	(0:2)		
RI	11	(5:6)					
Myotubes	GS Δ44	WT	SE	83	(51:32)		
			MXE	65	(36:29)		
			A5SS	9	(1:8)		
			A3SS	12	(5:7)		
RI	139	(8:131)					
Myotubes	GS Δ44	Δ44-1	SE	33	(10:23)		
			MXE	17	(9:8)		
			A5SS	5	(0:5)		
			A3SS	2	(0:2)		
RI	11	(5:6)					

SKIPPED EXON in GSA44 versus both WT and Δ44-1	
Myoblasts	
TFPI	tissue factor pathway inhibitor
ST7L	suppression of tumorigenicity 7 like
DCN	decorin
CD44	CD44 molecule (Indian blood group)
SLC4A7	solute carrier family 4, member 7
MAP4	microtubule-associated protein 4
PXN	paxillin
MYL6	myosin, light chain 6, smooth muscle and non-muscle
LGALS1	lectin, galactoside-binding, soluble, 1
ATP5SL	ATP5S-like
PRKAR1A	protein kinase, cAMP-dependent, type I, alpha
EIF4G2	eukaryotic translation initiation factor 4 gamma, 2
FN1	Fibronectin 1
CALD1	Caldesmon 1
RTN3	Reticulon 3
TANK	TRAF family member-associated NFKB activator
SULF1	Sulfatase 1
RPS24	ribosomal protein S24
PAM	peptidylglycine alpha-amidating monooxygenase
TMEM18	transmembrane protein 18
PLOD2	procollagen-lysine, 2-oxoglutarate 5-dioxygenase 2
COL6A3	collagen, type VI, alpha 3
USMG5	up-regulated during skeletal muscle growth 5 homolog
CD151	CD151 molecule (Raph blood group)
Myotubes	
CSDE1	cold shock domain containing E1, RNA-binding
TMED2	transmembrane emp24 domain trafficking protein 2
SULF1	Sulfatase 1
ITGB1	integrin, beta 1
TTN	titin

LEGEND: SE: Skipped exon
 MXE: Mutually exclusive exon
 A5SS: Alternative 5' splice site
 A3SS: Alternative 3' splice site
 RI: Retained intron

Supplementary Table 2. Alternative splicing analysis of WT, Δ44-1 and GSA44 cells.

Left panel shows the number of significant alternative splicing events in the indicated pairwise comparisons detected by MATS (by using both coverage of exons and spliced reads). The number of events is shown after the abbreviations explained in the legend. The right panel shows gene name and functional annotation of the genes marked by exon skipping events (SE) detected by MATS with FDR < 0.1 which differ concordantly between GSA44 and WT, GSA44 and Δ44-1, both in GM (top list) and DM (bottom list).

6) Figure 1 and 2: provide quantification for skipped bands in RT PCRs and Western blots.

The quantification of the skipped bands in Fig.1B and 2B is quite difficult since the unskipped bands are at saturation. More interesting is the fact that in both cases we recovered approximately 5% of dystrophin. The percentage of dystrophin expression with respect to control cells is approximately 5% and it is indicated on the top of the panel (the corresponding WT dilution was loaded in the same WB to avoid misleading quantification). The images shown in these figures are representative of multiple experiments, in all of them the percentage of dystrophin expression is always in the range of 5%. We have clarified this in the legends.

7) Karyotype analysis of newly generated iPSC lines would be required to match current international standards (eg <https://hpscereg.eu>). STR analysis would also be highly recommended to fingerprint iPSCs and parental fibroblasts.

As suggested by the reviewer we tried to contact Life & Brain in Bonn for this service but unfortunately, probably due to the Covid-19 emergency, they did not respond to our requests. Luckily, we were able to find a colleagues in Naples at the TIGEM institute who was able to perform cytogenetics analysis of the iPSCs that we raised. The results of such

analysis are presented in Fig EV1D and EV2D and show that all the karyotypes are normal.

8) Abstract: "inactivation of the splicing factor Celf2a was proven to be curative in other DMD backgrounds": the word curative is misleading, particularly for patients. Also, dystrophin is not rescued here, at best it would be partially rescued. Please tone down this statement.

We toned down this statement that has been modified in: "inactivation of the splicing factor Celf2a was proven to ameliorate the pathological state in other DMD backgrounds"

9) Opening introductory sentence: I would use "most" DMD mutations rather than "all".

The introductory sentence has been changed as suggested by the referee.

10) Are the mutations acronyms containing patients' initials? If that is the case this needs amending as might put patient confidentiality at risk.

GS Δ 44 have been already named in this manner in the previous paper, it would be a problem to change the nomenclature now. We have changed the name of the other Δ 44 line (UP Δ 44).

11) Please provide a diagram of the proposed model as final figure.

Following the reviewer's request we have added a proposed model in the final figure 5.

12) Title: I would argue that the mechanism shown in the manuscript is "associated with" rather than "drives" DMD amelioration.

We agree with the reviewer and we have modified the text.

13) The authors propose Celf2a ablation or inactivation as a novel therapeutic strategy for DMD. However, they should also discuss the feasibility and competitiveness of such a strategy with current approaches in clinical/preclinical trials. For example, it might be difficult to justify exposing (and immunizing) a DMD child to an AAV knocking out Celf2a by CRISPR and inducing only 5% dystrophin when the same AAV could induce higher dystrophin levels via permanent exon skipping or by delivering a microdystrophin.

We totally agree with the referee comments. The aim of the CRISPR/Cas9 approach, that we used in this study, was to demonstrate that the absence of Celf2a is able to partially rescue Dystrophin synthesis to obtain a "proof of concept" of our therapeutic strategy based on Celf2a inactivation. Our study opens the way to a possible drug treatment, which has to be preferred to a genetic strategy, for inactivating Celf2a activity. We have demonstrated in this work that the regulation of Celf2a is epigenetically mediated. The goal of our future project will be to set up epigenetic and pharmacological approaches that could allow the inhibition of Celf2a expression or activity. The final objective will be to obtain a drug treatments able to improve the quality of life, and possibly treat, almost 10% of DMD patients.

Referee #2 (Remarks for Author):

1) Figure 2A clearly shows that re-expression of Celf2a occurs during the process of myogenic differentiation of hiPSCs and not during the reprogramming of patient-derived fibroblasts into hiPSCs. The experiment shown in S2E indicates that MyoD and BAF60C overexpression in myocytes does not recover the expression of Celf2a. While this result formally excludes that Celf2a re-expression in GSdelta44 ipSC-derived myocytes is due to an artificial effect of the ectopic expression of MyoD and BAF60C in hiPSCs, it does not address the most important issue here - whether and Celf2a expression is a feature of hiPSC differentiation into somatic differentiated cells. While this reviewer understands that this is not a central aspect of the manuscript, the inclusion of this intriguing data in the manuscript prompts a number of questions of general cell biology and mechanistic insights. Not meaning to ask the authors to identify the mechanism responsible for this phenomenon, I still believe that it is important to define whether Celf2a re-expression is generally associated with hiPSC differentiation into specialized cell types.

To this purpose, I recommend that the author just validate Celf2a re-expression upon GSdelta44 ipSC differentiation into skeletal muscle, by using an alternative protocol with differentiation cues (such as those recently established in A. Pyle, O. Purquie' or G. Lee labs), and evaluate whether Celf2a re-expression occurs when GSdelta44 ipSC are induced to differentiate into at least one alternative lineages - e.g. neuronal, cardiomyocytes, hematopoietic, etc).

As already reported in answer to point 1 of Ref#1, the MHC immunofluorescence analysis and the measurements of the fusion index indicate that the different iPSC lines (control and mutants) are able to reach the myotube state. Moreover, as far as the protocol is concerned, we have previously experienced in Lenzi et al. (2016) two different differentiation procedures for converting iPSC into myotubes. In the first one only MyoD was employed and in this case terminal differentiation was reached at day 11 (dystrophin expression was used as a terminal differentiation marker). In the same paper we showed that the additional use of Baf60 allowed to shorten the differentiation process with dystrophin expression appearing at day 9. In this work, by using the MyoD/Baf60 procedure we get Celf2a expression at day 9 together with dystrophin (see Figure 1), while with the previous MyoD-only procedure Celf2a was visualized at day 11 (see Figure I in reply to Ref.#1).

Further prove that differentiation *per se* does not reactivate Celf2a derives from the following evidences: i) if you trans-differentiate fibroblasts (at the Weintraub manner), or myoblasts of GS Δ 44 into myotubes, Celf2a is never reactivated; ii) SFTA1P, which is an unrelated transcript originating from the same chromatin region of Celf2a and which is downregulated in GSD44, is instead re-expressed already in iPSCs. Therefore, it is a good control in order to prove that the epigenetic modification of the entire region has occurred during reprogramming, while the expression of Celf2a requires myogenic factors active only at later stages of differentiation.

Finally, in order to evaluate whether Celf2a re-expression could be generally due to hiPSC differentiation into somatic differentiated cells, we used a motoneuronal differentiation protocol already set up in our laboratory (De Santis et al. 2017). When WT iPSCs were differentiated to motor neuron progenitors (assessed by the expression of the motoneurons progenitor marker HB9). Celf2a was not turned on, suggesting that somatic differentiation *per se* is not the trigger for Celf2a re-expression (Figure IV).

FIGURE III. WT iPSCs differentiation to motoneurons progenitors (according to De Santis et al., 2017). The re-activation of *Celf2a* expression was analysed by RT-PCR in WT moroneuron progenitors

In conclusion, we show with two different procedures that *Celf2a* expression occurs in myogenic differentiation and it is not due to differentiation per se.

2) Combined ATACseq and ChIPseq analysis revealed the presence of potential regulatory elements at *Celf2a* locus; however, it is not clear whether they are promoters or enhancers, and whether they actually loop to regulate *Celf2a* promoter. Moreover, it is unclear whether changes in H3K27 and H3K4me3 signal, MyoD binding and chromatin accessibility, eventually influence these interactions. As it stands, this analysis remains quite inclusive.

In line with this request, we have strengthened the bioinformatics analysis of the *Celf2* locus. We have deepened our focus to the 4 ATAC-seq peaks that were found in the control condition (WT) and which were absent in *GSΔ44* (Figure 3A).

In first place, we clarify that all 4 ATAC-seq peaks are potential regulatory enhancers (not promoters). Indeed, all four peaks overlap with H3K27ac (modification typically associated to active enhancers) and MyoD peaks, while there is no correspondence with those of H3K4me3 (mark of active promoters; see Figure IV and Figure V here below). It is interesting to note that the H3K4me3 peaks present in *GSΔ44* map at the promoters of *Celf2b* and *c* whose expression is not affected in *GSΔ44*. Moreover, the gain of H3K27me3 and loss of H3K27ac at specific sites in *GSΔ44* parallels the chromatin compaction of the same regions and these two features correlate with the lack of *Celf2a* expression.

Figure IV. Genomic landscape of ATAC-seq peaks #1 and #2 (specific of WT).

Figure V. Genomic landscape of ATAC-seq peaks #3 and #4 (specific of WT).

Remarkably, we detected that the 4 ATAC-seq peaks (specific of WT) are located in blocks of evolutionary conservation, as identified by the PhastCons track of the UCSC (multi-species genomic alignments). Phylogenetic footprinting is indeed a potential marker of putative regulatory function.

To sum up, we have identified 4 regions along the Celf2 gene that present a distinct chromatin accessibility in WT Vs. GS Δ 44, together with characteristic epigenetic marks of active enhancers and ChIP-seq binding sites of MyoD.

Interestingly, no alteration of H3K4me3 peaks was observed in proximity of the Cef2b and c TSS in GS Δ 44 in line with the fact that both mRNAs are not affected in these conditions. Since these two mRNAs have a more ubiquitous expression, as indicated by their presence also in proliferating iPSCs (Figure 2a), it can be derived that the 4 ATAC peaks identified in the Celf2 locus are likely to correspond to myogenic-specific chromatin structures.

We have better explained all these observations in the text.

There are currently HiC maps available on high order chromatin interactions instigated by MyoD during activation of skeletal myogenesis in humans cells (Dall'Agnese et al. 2019) that could help capturing interactions between the regulatory elements identified in the Calf2a locus. The authors should use this resource and then exploit 3C analysis to experimentally monitor potential differences in these interactions between GSDelta44 myocytes and controls.

Following this suggestion, we have downloaded Hi-C data of the work by Dall'Agnese and colleagues on fibroblasts and myoblasts/myotubes derived from fibroblasts in Molecular Cell (2019). We found two sets of Hi-C differential interactions at 4Kb in the GEO entry GSE98530: E-GM vs. M-GM and M-GM vs. M-DM (E and M for Empty vector control and TET-inducible MyoD overexpression; GM and DM for growth medium and differentiation medium time points). From 63723 and 68064 interaction pairs respectively found at each file, we concentrated our interest in the interactions around the Celf2 gene (Figure VI).

Figure VI. Hi-C differential interactions in the region around the Celf2 gene during activation of skeletal myogenesis by Dall'Agnese et al (Molecular Cell 2019).

When focusing on this locus, we identified 3 pairs of M-GM vs. M-DM interactions in the genomic region encoding *Celf2* (Figure VII- green box). One interaction, that is lost in the DM timepoint (shown in blue, inside the green box) involves the TSS of *Celf2b* (left end) and a possible intronic enhancer (right end). This interaction includes one of our ATAC-seq peaks that are significantly lower in the $GS\Delta44$ condition (Figure VII). Moreover, this loop contains one of the MyoD ChIP-seq binding sites (shown in figure 3D and EV3D) that we demonstrated to be less accessible in $GS\Delta44$ cells. Interestingly, from our analysis, *Celf2b* is not muscle specific and does not need MyoD to be expressed (in fact it is present also in proliferating iPSC). Therefore, the occurrence of this loop in GM conditions could be needed for the expression of *CELF2a* and its formation could be inhibited in $GS\Delta44$ therefore preventing MyoD binding. Unfortunately, since we do not have Hi-C data for $GS\Delta44$ we cannot validate this hypothesis, but certainly this interesting observation will deserved attention for our future analysis.

3) I note that one major differential ATACseq peak between $GS\Delta44$ myocytes and controls locates within the *Celf2c* coding gene sequence and escaped the attention of the authors. This peak also coincides with MyoD peak and H3K27ac signal in WT myocytes that is lost in $GS\Delta44$ myocytes. I am wondering why the authors have not considered this as potential regulatory element.

We agree with the reviewer about the interest of this finding. In fact, this is the 3rd of our list of interesting ATAC-seq peaks (see Figure V), being located in an internal intron of the *Celf2* gene - therefore it is shared by the three isoforms. Interestingly, this peak is isolated from the rest of ATAC-seq peaks by the presence of two strong CTCF binding sites which could act as insulators that encapsulate the MyoD site within (Figure VII). The presence of intronic CTCF sites (according to ChIP-seq of HSMM -Skeletal Muscle Myoblast Cell- from the ENCODE project in the UCSC genome browser), which could act as insulators, suggests the existence of several insulators which could compartmentalize the *Celf2* gene

intronic region into three distinct areas, one them being the TSS of Celf2a/Celf2b and other the Celf2c transcript.

Figure VII. CTCF insulators in the Celf2 gene.

4) The relationship between expression of Celf2a and DUXAP8 is remain quite correlative in the absence of mechanistic evidence. Whether and how DUXAP8 regulates Celf2a by direct chromatin interaction at regulatory sequences (e.g. promoter, enhancers) is not addressed. I note from Fig. 3A that in GSDelta44 myocytes loss of H3K27Ac and H3K4me3, as well as reduction of chromatin accessibility, coincide with increased H3K27me3 signal, which reflects presence of active PRC2 complex. PRC2-mediated histone methylation has been associated with recruitment of regulatory lncRNA. EzH2 inhibition by soluble molecules or by siRNA-mediated downregulation should be used in conditions of DUXAP8 to determine whether it can counter the repressive effect on Celf2a. This evidence would at least indicate a potential mechanism of Celf2a repression by DUXAP8, via recruitment to a PRC2-bound regulatory element.

We agree that our data point to PRC2 repression mediated by DUXAP8. This RNA was previously shown to be associated with this complex, though in tissues different from muscle cells (Jiang et al., 2019; Gong et al., 2019; Lian et al., 2018; Ma et al., 2016). In order to check for this, we performed RIP analysis using Suz12 antibodies. As shown in the following figure, DUXAP8 was indeed found to interact to the PRC2 complex in agreement with what found in other tissues.

Figure VIII. The RNA level of DUXAP8 in SUZ12-immunoprecipitate was examined by qRT-PCR in WT mioblasts and presented as fold change compared with IgG-immunoprecipitate.

Unfortunately with the reagents available we were not able to perform RNAi against Ezh2 and analyze Celf2a expression. We got a plasmid expressing Ezh2 shRNA but unfortunately they were not effective in decreasing Ezh2 RNA levels. With the current contingency, the purchase of siRNAs is unlikely to occur in reasonable times. We hope that, since the DUXAP8 mechanism of action is not the central focus of the paper, the referee would not consider this experiment as an essential one for the conclusions we aim to draw.

Referee #3 (Remarks for Author):

1. In Figure S2C, it is unclear what the GSΔ44 DMD deletion means. My understanding is that the deletion of exon 44 is present only in GSΔ44 and not in GSM. What is the band present in GSΔ44 and GSM, but not in WT cells, in the GSΔ44 DMD deletion lane?

We have added a scheme since we agree that the gel can induce confusion. GSM (the mother of GS) is heterozygote for the D44 deletion; therefore she has both the band corresponding to the exon 44 amplification, as in the control, as well as the delta exon 44 product.

2. What is the H3K4me3, H3K27Ac and MyoD signal at the Celf2a genomic region upon reprogramming (iPSC) and differentiation (iPSC-derived myocytes)? Is there a rescue of H3K4me3, H3K27Ac, chromatin accessibility and MyoD enrichment at the Celf2a genomic region upon reprogramming and/or myogenic differentiation of iPSCs?

This is quite a challenging request since with iPSCs it is difficult to have enough material to perform all these analyses. Moreover, it is a very long experiment and with the ongoing Covid-19 restrictions it will take even longer. However, besides these intrinsic difficulties I think that we have sufficiently shown that the modulation occurs at the level of transcription and not at other levels (stability, translation or other).

3. DUXAP8 is highly homologous to the protein-coding gene DUXA and to other DUXA-related transcripts. It is therefore essential that the authors demonstrate that their reagents are DUXAP8-specific. For example, are the primers used to detect DUXAP8 selective for this transcript? Moreover, expression of DUXA and the other DUXA-related pseudogenes must be shown in all instances in which DUXAP8 is manipulated to demonstrate DUXAP8 siRNAs and OE are not affecting the expression of DUXA and the other DUXA-related pseudogenes.

The reagents that were used do not recognize DUXA. As shown in Figure IX the region of identity between DUXAP8 and DUXA is located in DUXAP8 in the exon 8. The region recognized by the siRNA that was used in this study (represented by a red line in the picture) is located in the exon 7, which is not present in DUXA. Moreover, the PCR oligonucleotides that we used do not amplify DUXA (blue and green arrows). Blast analysis and in silico PCR also confirmed the specificity.

Figure IX. DUXAP8 representative scheme.

There are two transcripts related to the DUXA pseudogene (DUXAP9 and DUXAP10) that cannot be distinguished, neither by PCR nor by siRNA, from DUXAP8 because they share 99.6% gDNA identity. We refer our results to DUXAP8 gene in line with existing papers from other groups that even if referring to DUXAP8, used PCR oligonucleotides that were able to recognize also DUXAP9 and 10 (Sun et al., 2017; Ma et al., 2016; Lian et al., 2018; Jiang et al., 2019).

Which is the physiological expression pattern of DUXAP8? In which human tissue is DUXAP8 normally expressed? Is the expression of DUXAP8 and Celf2a physiologically also mutually exclusive? In other words, is DUXAP8 expressed only in pathological conditions or is DUXAP8 involved in the regulation of Celf2a expression in normal physiology?

DUXAP8 is normally expressed during development, and down-regulated after birth. Therefore, in muscles the lack of its expression correlates well with Celf2a expression in physiological conditions. Though there are conditions in which DUXAP8 was found overexpressed in adult tissues which specifically relate to cancerous states (one example among many is epatocellular carcinoma - Wei et al., 2020). We have analyzed some of those data (NCBI GENE website, <https://www.ncbi.nlm.nih.gov/gene/503637>) but in no case it was possible to correlate DUXAP8 expression with that of Celf2a since the data available do not distinguish between the Celf2 isoforms (a, b and c) but they only examine the global expression of CELF2 as the sum of all three isoforms (<https://www.ncbi.nlm.nih.gov/gene/10659>).

Therefore, the re-expression of DUXAP8 in GS Δ 44 myoblasts is a very interesting issue that should be addressed also to understand the genetic asset of GS Δ 44.

4. There are a number of mechanistic aspects that must be addressed.

What is the mechanism responsible for the differential DUXAP8 expression between the

GSM, GSF and GS Δ 44 and between the primary fibroblasts and the iPSCs and iPSC-derived myocytes? Are the ChIP-seq and ATAC-seq data available to the authors showing any significant change at the DUXAP8 locus?

What is the mechanism by which DUXAP8 regulate Celf2a expression? Is it direct? Is DUXAP8 associated to the Celf2a genomic region? DUXAP8 has been reported to regulate gene expression by positively regulating PRC2 and LSD1, and by sponging miR-422a and miR-577. Are PRC2 and/or LSD1 differentially enriched at the Celf2a locus in WT vs GS Δ 44, siSCR vs siDUXAP8, and in Empty vs OE DUXAP8 cells? Is Celf2a a miR-422a and/or miR-577 target? no effect on Celf2a

Unfortunately, from the available ChIP-seq and ATAC-seq data, the analysis of DUXAP8 locus was not possible, probably due to low coverage/number of reads and to the repetitive nature of the sequence itself.

Following the suggestion of this reviewer (see also the reply to reviewer #1, point 2) we reanalyzed the sequencing profile of H3K27me3 (the epigenetic product of PRC2) on the genomic region that contains Celf2a. As it is shown in Figure X, the Celf2a regulatory enhancers containing MyoD binding sites are decorated by two peaks of H3K27me3 precisely in the GS Δ 44 condition (highlighted in blue – Figure X), while the same enhancers were not marked by H3K27me3 in the control condition (WT). H3K27me3 is typically associated to gene silencing by Polycomb.

Remarkably, no peaks are observed when examining the ChIP-seq of H3K27me3 in HSMM (Skeletal Muscle Myoblast Cell) by the ENCODE project.

To sum up, we consider that these differences in H3K27me3 levels are significant and that, therefore, it is tempting to speculate that PRC2 could mediate the regulation/repression of Celf2a via DUXAP8 in GS Δ 44, being associated to a compaction of the chromatin in the same regulatory regions.

A comprehensive search involving multiple bioinformatics resources of the known targets of the DUXAP8 lncRNA (lncRNA2target, lncRRISearch, lncTarD) was performed and even if no known interaction between DUXAP8 and Celf2 has been reported until now, this possibility cannot be excluded.

Figure X. Profiles of H3K27me3 in WT and GS Δ 44 and H3K27me3/EZH2 in HSMM on Celf2a

Several papers from other groups have shown, by RIP experiments, that different PRC2 component are able to interact with DUXAP8 (Jiang et al., 2019; Gong et al., 2019; Lian et al., 2018; Ma et al., 2016). In order to check whether this is true also in our system, we performed RIP analysis using Suz12 antibodies. As shown in the Figure VIII (see at p. 10) DUXAP8 was indeed found to interact to the PRC2 complex in agreement with what found in other tissues.

Finally, the hypothesis of miRNAs mediated regulation was excluded by the observation that the three Celf2 isoforms share the same 3' UTR, while only the expression of Celf2a is affected in our patient. Moreover, there are no predicted binding sites for miR-422a in the Celf2 3'UTR while only two "poorly conserved" binding sites for miR-577, with a very low score were detected (Figure XI, according to Targetscan predictor - Agarwal V, Bell GW, Nam J, Bartel DP. Predicting effective microRNA target sites in mammalian mRNAs. *eLife*, 4:e05005, (2015). *eLife* Lens view). A recent paper (Ji et al., 2020) identified DUXAP8 as sponge for miR-498, but also in this case there are only two predicted "poorly conserved" binding sites for this miR in the 3' UTR of Celf2.

Figure XI. miR-577 and miR-498 predicted binding sites on Celf2 3' UTR using Targetscan predictor.

Poorly conserved

	Predicted consequential pairing of target region (top) and miRNA (bottom)	Site type	Context++ score	Context++ score percentile	Weighted context++ score	Conserved branch length	P _{CT}
Position 760-766 of CELF2 3' UTR hsa-miR-498	5' ...AAAUACUUGAUGUUUCUUGAAAG... 3' CUUUUUGCGGGGACCGAACUUU	7mer-A1	-0.03	48	-0.01	0.043	N/A
Position 1718-1724 of CELF2 3' UTR hsa-miR-498	5' ...UUUUCUAAAACAGACUUGAAAG... 3' CUUUUUGCGGGGACCGAACUUU	7mer-A1	-0.02	48	-0.01	0.043	N/A

Poorly conserved

	Predicted consequential pairing of target region (top) and miRNA (bottom)	Site type	Context++ score	Context++ score percentile	Weighted context++ score	Conserved branch length	P _{CT}
Position 1093-1099 of CELF2 3' UTR hsa-miR-577	5' ...UGUUUGUAACUCCAUUAUCUAG... 3' GUCCAUGGUUUAUAAAUAGAU	7mer-A1	-0.03	61	-0.03	0.043	N/A
Position 4228-4234 of CELF2 3' UTR hsa-miR-577	5' ...GUCCUUCUACCACU--UUUAUCUU... 3' GUCCAUGGUUUAUAAAUAGAU	7mer-m8	-0.05	76	-0.05	0.043	N/A

In conclusion, our results indicate that a PRC2 signature is found on the Celf2a locus and that is dependent on the presence of DUXAP8. Moreover, we confirm that also in our system DUXAP8 is bound to a component of the PRC2 complex. Further work will elucidate whether DUXAP8 is directly interacting with the Celf2a locus or whether its effect is indirect. Considering the complexity of such analysis we hope that the reviewer will agree that this study should be the subject of future work.

28th May 2020

Dear Prof. Bozzoni,

Thank you for the submission of your revised manuscript to EMBO Molecular Medicine and apologies for the delay in getting back to you. As you will see from the set of comments pasted below, I needed to cross-comment once more prior to making a decision. We have now received the enclosed reports from the referees who were asked to re-assess it. As you will see, while referees #1 and #3 are now supportive, referee #2 remains concerned by the provision of limited mechanism. Therefore, we would like to propose the following:

1) You will see that referee #2 remains unsatisfied with the data and would very much like to see more causative experiments. This referee recommends a course of action to perform additional epigenetic analysis that would ascertain the role of DUXAP8 on Celf2a expression in the context of GSDelta44 genetic background. During cross-commenting, referee #3 agreed that such experiments would be valuable to the study and really improve its conclusiveness. Still, we understand that in the current context of COVID-19 restrictions, such experiments would mean another 3-6 months work. As such, we decided that we will not ask you to perform these experiments but we would ask that you make it crystal clear in the main article that the mechanism remains correlative and tone down the conclusions of the paper accordingly.

Please provide a point-by-point letter INCLUDING my comments as well as the reviewer's reports and your detailed responses to their comments (as Word file).

2) Source Data:

We encourage the publication of source data, particularly for electrophoretic gels, blots, but also microscopy images with the aim of making primary data more accessible and transparent to the reader. Would you be willing to provide a PDF file per figure that contains the original, uncropped and unprocessed scans of all or key gels used in the figure (including molecular weight markers)? The PDF files should be labeled with the appropriate figure/panel number (1 file/figure), and should have molecular weight markers; further annotation may be useful but is not essential. The PDF files will be published online with the article as supplementary "Source Data" files. If you have any questions regarding this just contact me.

Please note that the controls used in figures 2A and EV4F are the same. Please provide an explanation and update the legends accordingly.

3) In the main manuscript file, please do the following:

- correct/answer the track changes suggested by our data editors by working from the attached/uploaded document
- Add the legend for Table EV1 within the file and remove it from the main article document
- in M&M, the statistical paragraph should reflect all information that you have filled in the Authors checklist, especially regarding randomisation, blinding, replication.
- indicate in legends exact $n=$ and exact $p=$ values, not a range, along with the statistical test used. Some people found that to keep the figures clear, providing an Appendix table Sx with all exact p -values was preferable. You are welcome to do this if you want to.
- in M&M, include a statement that informed consent was obtained from all human subjects and

that the experiments conformed to the principles set out in the WMA Declaration of Helsinki and the Department of Health and Human Services Belmont Report [even if the cells were published before].

4) As part of the EMBO Publications transparent editorial process initiative (see our Editorial at <http://embomolmed.embopress.org/content/2/9/329>), EMBO Molecular Medicine will publish online a Review Process File (RPF) to accompany accepted manuscripts.

In the event of acceptance, this file will be published in conjunction with your paper and will include the anonymous referee reports, your point-by-point response and all pertinent correspondence relating to the manuscript. Let us know whether you agree with the publication of the RPF and as here, if you want to remove or not any figures from it prior to publication.

5) Data availability section.

I have updated the text according to our guidelines, please double check that it's all correct and make sure to request that the data is made available as soon as the paper will be accepted, and not on Oct 28, 2020.

Please submit your revised manuscript within two weeks. I look forward to seeing a revised form of your manuscript as soon as possible.

Yours sincerely,

Celine Carret

Celine Carret, PhD
Senior Editor
EMBO Molecular Medicine

*** Instructions to submit your revised manuscript ***

To submit your manuscript, please follow this link:

<https://embomolmed.msubmit.net/cgi-bin/main.plex>

***** Reviewer's comments *****

Referee #1 (Remarks for Author):

The authors have addressed most of my points and I am satisfied with the revised version of the manuscript, which has overall improved.

Referee #2 (Comments on Novelty/Model System for Author):

It is still unclear how iPSC-mediated differentiation into skeletal muscles is the optimal experimental model to address the function of Celf2a, given that Celf2a expression changes during reprogramming into iPSCs and their differentiation into muscle

Referee #2 (Remarks for Author):

Although the authors have been quite responsive to the criticisms raised by reviewers, there are serious concerns remaining on this manuscript, as outlined below.

It remains unclear the mechanistic basis and biological meaning of Celf2a re-expression during GSdelta44 iPSC differentiation into skeletal muscles. The author shows that GSdelta44 patient and mother's fibroblasts are devoid of Celf2a expression; however, Celf2a expression is recovered upon fibroblasts reprogramming into iPSCs and then into differentiated muscle cells. By contrast, direct myogenic conversion of GSdelta44 fibroblast by ectopic expression of MyoD results in formation of skeletal muscles that do not express Celf2a. While this difference would suggest that reprogramming of GSdelta44 patient-derived fibroblasts into iPSCs is associated with the re-expression of Celf2a, Fig. 2A shows that GSdelta44 iPSCs do not express Celf2a, but Celf2a expression is resumed during GSdelta44 iPSC differentiation into skeletal muscles. Curiously, in the same figure the authors show that also WT#1 iPSCs do not express Celf2a, indicating that Celf2a silencing could be a general feature of somatic cell reprogramming into iPSCs and that iPSC-differentiation could de-repress Celf2a expression.

Unfortunately, the authors did not clarify this point, and it remains unclear whether: a) Celf2a expression is affected by somatic cell reprogramming into iPSCs or by iPSC-differentiation into skeletal muscle; b) re-activation of Celf2a expression is a specific feature of GSdelta44 patient-derived iPSC differentiation into skeletal muscles; c) Silencing of Celf2a is a general feature of somatic cell reprogramming into iPSCs and re-activation of Celf2a is a general feature of iPSC differentiation into skeletal muscles.

The data shown by the authors suggest that loss of Celf2a expression is a general feature of somatic cell reprogramming into iPSCs and that Celf2a is resumed upon their differentiation into skeletal muscles. This is in conflict with the sentence in the second paragraph of the manuscript "... the abrogation of the Celf2a factor in the GSdelta44 case is due to epigenetic repression that is lost upon reprogramming into iPSCs". This sentence is also incorrect in principle, as reprogramming into iPSCs does not reactivate Celf2a expression (as shown in Fig. 2A).

I think the authors need to absolutely clarify whether and why Celf2a expression is lost during normal fibroblast reprogramming in iPSCs, and whether Celf2a de-repression is a specific feature of iPSC-differentiation into skeletal muscle cells. If this is the case, the authors should determine whether there is an inverse correlation of DUXAP8 levels also in the regulation of Celf2a during normal fibroblast reprogramming in iPSCs, as it is in the case of GSdelta44 patient-derived cells. Finally, the authors did not properly address my initial concern regarding the relationship between Celf2a re-expression and iPSC differentiation into skeletal muscle or other lineages. I therefore reiterate my previous recommendation of validating Celf2a in GSdelta44 iPSC upon differentiation into skeletal muscle, by using an alternative protocol with differentiation cues (such as those recently established in A. Pyle, O. Purquie' or G. Lee labs), as well as upon their differentiation into at least one alternative lineage - e.g. motoneurons).

Referee #3 (Remarks for Author):

While it would have been desirable to have more mechanistic insights concerning the regulation of DUXAP8 expression and how it regulates Celf2a expression, the authors have sufficiently addressed my concerns.

Dear Dr Carret,

thank you for your response. We appreciate that you would not require further experiments but that we better underline the fact that the mechanism of Celf2a regulation by DUXAP8 is still correlative.

We have carefully checked throughout the manuscript those parts where this could have happened. Indeed, we are the first to admit that for DUXAP8 only a correlative link exists with Celf2a repression. Instead, the take home message we wanted to send was that, for a relevant genetic disease, an inherited epigenetic repression is abolished upon reprogramming. Moreover, we have better clarified that, since Celf2a has a tissue-specific expression mainly restricted to muscle, you need to induce myogenic differentiation of iPSCs in order to recover transcription of the gene.

These aspects have been better clarified in the text (highlighted in yellow) and in response to ref.#2 (see below).

Referee #2 (Comments on Novelty/Model System for Author):

It is still unclear how iPSC-mediated differentiation into skeletal muscles is the optimal experimental model to address the function of Celf2a, given that Celf2a expression changes during reprogramming into iPSCs and their differentiation into muscle

Probably we were not clear enough on this point, but the Celf2a isoform has a tissue-specific expression mainly restricted to muscle cells. This is why in order to study its expression we have to induce myogenic differentiation of iPSCs.

Referee #2 (Remarks for Author):

Although the authors have been quite responsive to the criticisms raised by reviewers, there are serious concerns remaining on this manuscript, as outlined below.

It remains unclear the mechanistic basis and biological meaning of Celf2a re-expression during GSdelta44 iPSC differentiation into skeletal muscles. The author show that GSdelta44 patient and mother's fibroblasts are devoid of Celf2a expression; however, Celf2a expression is recovered upon fibroblasts reprogramming into iPSCs and then into differentiated muscle cells. By contrast, direct myogenic conversion of GSdelta44 fibroblast by ectopic expression of MyoD results in formation of skeletal muscles that do not express Celf2a. While this difference would suggest that re-programming of GSdelta44 patient-derived fibroblasts into iPSCs is associated with the re-expression of Celf2a, Fig. 2A shows that GSdelta44 iPSCs do not express Celf2a, but Celf2a expression is resumed during GSdelta44 iPSC differentiation into skeletal muscles. Curiously, in the same figure the authors show that also WT#1 iPSCs do not express Celf2a, indicating that Celf2a silencing could be a general feature of somatic cell reprogramming into iPSCs and that iPSC-differentiation could de-repress Celf2a expression.

We apologize for not being clear enough; here below we summarize the main features of Celf2a regulation:

- 1) The Celf2a is an isoform which is **specifically expressed in muscle cells** and which responds to myogenic-specific transcriptional activators (the master myogenic regulator MyoD being the best candidate, as shown by the ChIP data presented in the paper). Celf2a is neither expressed in control nor in GSΔ44

fibroblasts and iPSCs. Therefore, you need to have myogenic cells in order to detect its expression. This has been better clarified in the text (see the parts highlighted in yellow).

- 2) Celf2a expression is under epigenetic control: ATAC-seq analyses performed in control and GS Δ 44 myoblasts clearly demonstrate that the chromatin region containing Celf2a is closed in myoblasts derived from the patient while it is accessible in controls. This suggests that even in the presence of protein factors that are necessary for its transcription, Celf2a is not produced in GS Δ 44 myoblasts because of the repressive state of its chromatin.
- 3) Direct trans-differentiation **does not** allow to remove epigenetic repression from the Celf2a locus: while control fibroblasts, when trans-differentiated into myocytes, correctly express Celf2a, this does not occur for GS Δ 44 indicating that in these cells the Celf2a locus is already repressed and remains repressed upon differentiation.
- 4) At variance with direct trans-differentiation, reprogramming **does** allow to remove epigenetic repression: when the same GS Δ 44 fibroblasts are first reprogrammed into iPSCs and then converted into myocytes, Celf2a expression is recovered. Note that the chromatin state of the chromosomal region containing the Celf2a locus is indeed open in GS Δ 44 iPSCs as indicated by the reactivation of the neighboring SFTA1P gene (Extended data Fig.4F).

Therefore, reprogramming and not direct trans-differentiation, allows the removal of the epigenetic repression of Celf2a. However, this is only the first step since you then need to induce myogenic differentiation in order to transcribe the locus. Control and mutant iPSCs do not express Celf2a (**Fig. 2A**) because they are devoid of specific myogenic transcriptional activators.

In conclusion, the reactivation of the Celf2a locus is a two step process (see sketch below): the first includes the removal of epigenetic repression during reprogramming, while the second requires differentiation into skeletal muscle in order to have the transcriptional factors required for the tissue -specific activation of the gene.

GS Δ 44 Myoblasts

Closed chromatin

GS Δ 44 iPSCs

Permissive chromatin

GS Δ 44 iPSCs differentiation into myotubes

Permissive chromatin and tissue-specific transcription factors that allow the production of the Celf2a.

Unfortunately, the authors did not clarify this point, and it remains unclear whether:

a) Celf2a expression is affected by somatic cell reprogramming into iPSCs or by iPSC-differentiation into skeletal muscle;

For the above reasons we believe that the expression of Celf2a is allowed by a two step process: the first is the permissive state of the chromatin established by reprogramming, while the second is the transcriptional activation by muscle specific transcriptional factors in myogenic cells.

b) re-activation of Celf2a expression is a specific feature of GSdelta44 patient-derived iPSC differentiation into skeletal muscles;

Yes, Celf2a re-activation only occurs in myocytes derived from reprogrammed fibroblasts and not from trans-differentiated ones.

c) Silencing of Celf2a is a general feature of somatic cell reprogramming into iPSCs and re-activation of Celf2a is a general feature of iPSC differentiation into skeletal muscles.

No, Celf2a is not silenced during somatic cell reprogramming. iPSCs do not express Celf2a because they lack the TFs needed for its activation. Indeed, the chromatin state of the Celf2a genomic region in iPSCs is permissive as demonstrated by the re-expression of the neighboring SFTA1P gene (Extended data Fig.4F). This transcript is repressed similarly to Celf2a in GSD44 myoblasts but, differently from Celf2a, it does not require myogenic transcriptional activators.

The data shown by the authors suggest that loss of Celf2a expression is a general feature of somatic cell reprogramming into iPSCs and that Celf2a is resumed upon their differentiation into skeletal muscles. This is in conflict with the sentence in the second paragraph of the manuscript "... the abrogation of the Celf2a factor in the GSdelta44 case is due to epigenetic repression that is lost upon reprogramming into iPSCs". This sentence is also incorrect in principle, as reprogramming into iPSCs does not reactivate Celf2a expression (as shown in Fig. 2A).

Reprogramming into iPSCs does not reactivate Celf2a expression but allows the remodeling of the chromatin that acquires a permissive state that ensures its expression when the needed TFs are present. We have rephrased the sentence accordingly.

I think the authors need to absolutely clarify whether and why Celf2a expression is lost during normal fibroblast reprogramming in iPSCs, and whether Celf2a de-repression is a specific feature of iPSC-differentiation into skeletal muscle cells. If this is the case, the authors should determine whether there is an inverse correlation of DUXAP8 levels also in the regulation of Celf2a during normal fibroblast reprogramming in iPSCs, as it is in the case of GSdelta44 patient-derived cells.

Along the direction of DUXAP8 controlling the chromatin state of the Celf2a locus, the correlation is quite good since DUXAP8 is not expressed in reprogrammed GS Δ 44 iPSCs (Fig.4E) while present in fibroblasts (Fig.4D).

Finally, the authors did not properly address my initial concern regarding the relationship between Celf2a re-expression and iPSC differentiation into skeletal muscle or other lineages. I therefore reiterate my previous recommendation of validating Celf2a in GSdelta44 iPSC upon differentiation into skeletal muscle, by using an alternative protocol with differentiation cues (such as those recently established in A. Pyle, O. Purquie' or G. Lee labs), as well as upon their differentiation into at least one alternative lineages - e.g. motoneurons).

We used two slightly different procedures for inducing myogenic differentiation and in both cases expression of bona fide markers of terminal muscle differentiation was obtained (Myosin Heavy Chain and Dystrophin). We believe that this should reassure about the fidelity of the system utilized.

Your comments:

1) You will see that referee #2 remains unsatisfied with the data and would very much like to see more causative experiments. This referee recommends a course of action to perform additional epigenetic analysis that would ascertain the role of DUXAP8 on Celf2a expression in the context of GSdelta44 genetic background. During cross-commenting, referee #2 agreed that such experiments would be valuable to the study and really improve its conclusiveness. Still, we understand that in the current context of COVID-19 restrictions, such experiments would mean another 3-6 months work. As such, we decided that we will not ask you to perform these experiments but we would ask that you make it crystal clear in the main article that the mechanism remains correlative and tone down the conclusions of the paper accordingly.

See above

Please provide a point-by-point letter INCLUDING my comments as well as the reviewer's reports and your detailed responses to their comments (as Word file).

2) Source Data:

We encourage the publication of source data, particularly for electrophoretic gels, blots, but also microscopy images with the aim of making primary data more accessible and transparent to the reader. Would you be willing to provide a PDF file per figure that contains the original, uncropped and unprocessed scans of all or key gels used in the figure (including molecular weight markers)? The PDF files should be labeled with the appropriate figure/panel number (1 file/figure), and should have molecular weight markers; further annotation may be useful but is not essential. The PDF files will be published online with the article as supplementary "Source Data" files. If you have any questions regarding this just contact me.

Pdf files containing "Source Data" have been added as suggested

Please note that the controls used in figures 2A and EV4F are the same. Please provide an explanation and update the legends accordingly.

This problem has been solved by appropriately modifying the legend of the figure EV4F.

3) In the main manuscript file, please do the following:

- correct/answer the track changes suggested by our data editors by working from the attached/uploaded document
- Add the legend for Table EV1 within the file and remove it from the main article document
- in M&M, the statistical paragraph should reflect all information that you have filled in the Authors checklist, especially regarding randomisation, blinding, replication.
- indicate in legends exact n= and exact p= values, not a range, along with the statistical test used. Some people found that to keep the figures clear, providing an Appendix table Sx with all exact p-values was preferable. You are welcome to do this if you want to.
- in M&M, include a statement that informed consent was obtained from all human subjects and that the experiments conformed to the principles set out in the WMA Declaration of Helsinki and the Department of Health and Human Services Belmont Report [even if the cells were published before].

We've done all the suggested actions

4) As part of the EMBO Publications transparent editorial process initiative (see our Editorial at <http://embomolmed.embopress.org/content/2/9/329>), EMBO Molecular Medicine will publish online a Review Process File (RPF) to accompany accepted manuscripts.

In the event of acceptance, this file will be published in conjunction with your paper and will include the anonymous referee reports, your point-by-point response and all pertinent correspondence relating to the manuscript. Let us know whether you agree with the publication of the RPF and as here, if you want to remove or not any figures from it prior to publication.

I agree with the publication of the RPF.

5) Data availability section.

I have updated the text according to our guidelines, please double check that it's all correct and make sure to request that the data is made available as soon as the paper will be accepted, and not on Oct 28, 2020.

I have checked, everything is correct and the data will be available as soon as the paper will be accepted.

Please submit your revised manuscript within two weeks. I look forward to seeing a revised form of your manuscript as soon as possible.

Yours sincerely,

Celine Carret

Celine Carret, PhD

Senior Editor
EMBO Molecular Medicine

*** Instructions to submit your revised manuscript ***

To submit your manuscript, please follow this link:

<https://embomolmed.msubmit.net/cgi-bin/main.plex>

***** Reviewer's comments *****

Referee #1 (Remarks for Author):

The authors have addressed most of my points and I am satisfied with the revised version of the manuscript, which has overall improved.

Referee #3 (Remarks for Author):

While it would have been desirable to have more mechanistic insights concerning the regulation of DUXAP8 expression and how it regulates Celf2a expression, the authors have sufficiently addressed my concerns.

Yours sincerely,

Irene Bozzoni

8th Jun 2020

Dear Prof. Bozzoni,

Thank you for submitting your revised article. We are pleased to inform you that your manuscript is accepted for publication and is now being sent to our publisher to be included in the next available issue of EMBO Molecular Medicine.

Please read below for additional IMPORTANT information regarding your article, its publication and the production process.

Congratulations on your interesting work,

Celine

Celine Carret, PhD
Senior Editor
EMBO Molecular Medicine

Follow us on Twitter @EmboMolMed
Sign up for eTOCs at embopress.org/alertsfeeds

Corresponding Author Name: Irene Bozzoni

Journal Submitted to: EMBO MOLECULAR MEDICINE

Manuscript Number: EMM-2020-12063-T